# YAP is essential for mechanical force production and epithelial cell proliferation during lung branching morphogenesis

Chuwen Lin[1], Erica Yao[1], Kuan Zhang[1], Xuan Jiang[1], Stacey Croll[1], Katherine Thompson-Peer[2], Pao-Tien Chuang[1]*

[1]Cardiovascular Research Institute, University of California, San Francisco, San Francisco, United States; [2]Department of Physiology, Howard Hughes Medical institute, University of California, San Francisco, San Francisco, United States

**Abstract** Branching morphogenesis is a fundamental program for tissue patterning. We show that active YAP, a key mediator of Hippo signaling, is distributed throughout the murine lung epithelium and loss of epithelial YAP severely disrupts branching. Failure to branch is restricted to regions where YAP activity is removed. This suggests that YAP controls local epithelial cell properties. In support of this model, mechanical force production is compromised and cell proliferation is reduced in *Yap* mutant lungs. We propose that defective force generation and insufficient epithelial cell number underlie the branching defects. Through genomic analysis, we also uncovered a feedback control of pMLC levels, which is critical for mechanical force production, likely through the direct induction of multiple regulators by YAP. Our work provides a molecular pathway that could control epithelial cell properties required for proper morphogenetic movement and pattern formation.

*For correspondence: pao-tien. chuang@ucsf.edu

**Competing interests:** The authors declare that no competing interests exist.

## Introduction

Development of the lung has served as a model system to study fundamental questions such as branching morphogenesis, epithelial-mesenchymal interactions and cell-type specification. The lung primordium arises from the ventral foregut and its emergence is marked by the expression of the *Nkx2.1* transcription factor (*Zhou et al., 1996*; *Minoo et al., 1999*). The lung primordium is composed of two parts: the future trachea and two endodermal buds. Both components are composed of an epithelial layer of endoderm surrounded by mesodermal cells. During lung branching morphogenesis, three characteristic modes of branching are repeatedly used at many different times and positions (*Metzger et al., 2008*). They include formation of lateral branches from the parent branch (domain branching) and bifurcation at the tip of branches (planar and orthogonal bifurcation) (*Metzger et al., 2008*). Initially, the buds grow ventrally and caudally, and initiate lateral branches at invariant positions, beginning around 10.5 *days post coitus* (*dpc*) in mice. In this way, five buds are generated, four on the right side and one on the left side, leading to the formation of four right lobes (cranial, middle, accessory and caudal) and one left lobe of the mature mouse lung. In this process, proximal-distal specification is associated with the emergence of *Sox9*-expressing progenitors and non-branching *Sox2*-expressing airways, which will become the conducting airway in the mature lung (*Yang and Chen, 2014*).

These complicated morphological movements culminate in an elaborate branching organ that also contains various differentiated cell types to fulfill the primary function of gas exchange. Acquisition of the spatial and temporal sequence of lung morphogenesis forms the basis of understanding the molecular mechanisms of lung development.

**eLife digest** Air enters our lungs through a system of airways that spread outwards from the windpipe like the branches of a tree. Before we are born, each branch is shaped by the organization and movement of cells that form the walls of the airways, called epithelial cells. This process requires the cells to communicate and coordinate with each other by receiving and/or sending chemical signals.

One important system that epithelial cells use to communicate is called the Hippo pathway, which uses a molecule called YAP to execute received messages. Exactly how YAP helps airways in the lungs to develop was not well understood.

By studying developing mouse lungs, Lin et al. have now found that YAP is present in the epithelial cells of all developing airways. Inactivating YAP in specific parts of the lungs prevented the formation of new branches of the airway in just those regions that lacked YAP. This suggests that YAP is needed for airways to branch properly and form the extensive network present in healthy lungs.

Further investigation using genomics approaches revealed that YAP regulates the activity of genes that control how epithelial cells divide and contract. Without YAP, fewer cells were produced and they were unable to produce the forces required to change shape and move to form airways. In particular, YAP controls the production of a modified form of a protein called phosphorylated myosin light chain (pMLC) through a regulatory pathway. The pMLC protein is critical for the cells to produce the mechanical forces that they need to be able to contract correctly.

Overall, the results presented by Lin et al. suggest that YAP controls the properties of the epithelial cells to enable them to form new airway branches. Branched structures also form in a number of other organs, and the mechanisms that cause these structures to form are thought to be similar to those that form the airways. Lin et al.'s work could therefore help us to understand how organs develop more generally.

Precise control of lung growth and patterning is essential for the generation of a functional respiratory system. Several genes and pathways involved in this process have been identified (*Morrisey et al., 2013*; *Ornitz and Yin, 2012*; *Volckaert et al., 2015*; *Domyan and Sun, 2011*). However, we lack a mechanistic understanding of how lung epithelial cell properties are dictated by these genes and pathways during growth and patterning. This is a central unresolved issue in understanding lung development and homeostasis. Since lung growth and patterning is a tightly controlled process, we speculate that a master regulatory pathway is required for integrating diverse inputs and coordinating a multitude of cellular behaviors. A strong candidate is the Hippo pathway, which mediates organ size control in several other organs. Indeed, several recent studies show that the Hippo pathway controls key aspects of lung development, homeostasis and repair (*Mahoney et al., 2014*; *Zhao et al., 2014*; *Lange et al., 2015*; *Lin et al., 2015*).

Extensive work in cell-based assays and model organisms has culminated in a working model of Hippo signaling (*Yu and Guan, 2013*; *Pan, 2010*; *Gumbiner and Kim, 2014*; *Matsui and Lai, 2013*; *Harvey and Hariharan, 2012*; *Barry and Camargo, 2013*; *Staley and Irvine, 2012*; *Halder and Johnson, 2011*). The key executor of the Hippo pathway is the transcriptional coactivator Yes-associated protein (YAP), which governs multiple aspects of cell physiology (*Varelas, 2014*). YAP activity is negatively regulated through a kinase cascade, and YAP phosphorylation is correlated with its cytoplasmic sequestration and degradation. Consequently, the transcriptional targets of YAP in the nucleus are not expressed. In contrast, when upstream kinases are inactivated in response to external signals (such as low cell density), YAP becomes hypophosphorylated. It is postulated that this form of YAP enters the nucleus and activates Hippo targets in conjunction with the TEAD1-4 transcription factors. This is required for orchestrating a multitude of cellular functions including cell proliferation, differentiation and death and changes in cell properties.

In this study, we utilize the Hippo pathway as a tool to gain insight into the molecular mechanisms of lung development. Our findings support a model in which control of lung epithelial cell properties, such as cell proliferation and mechanical force production, by Hippo signaling determines lung

growth and patterning. This knowledge significantly enhances our mechanistic understanding of lung branching morphogenesis.

## Results

### Active nuclear YAP is distributed throughout the airway epithelium and is not confined to a specific zone

To assess the sites of active (nuclear) YAP at different stages of lung development, we examined the subcellular distribution of YAP protein in the lung epithelium by immunofluorescence and immuno-histochemistry. Nuclear YAP could be found in both SOX2$^+$ (proximal) and SOX9$^+$ (distal) epithelial populations in wild-type lungs at 11.5, 12.5 and 14.5 *dpc* (*Figure 1A–P*; *Figure 1—figure supplement 1*), suggesting that YAP is active throughout the lung epithelium. YAP staining was barely detectable in the epithelium but was present at wild-type levels in the mesenchyme of *Yap*-deficient (*Yap$^{f/f}$; Shh$^{Cre/+}$*) lungs (see below) (*Figure 1Q,R*; *Figure 1—figure supplement 1M*), indicating the specificity of YAP antibodies. A high percentage of SOX2$^+$ cells and SOX9$^+$ cells displayed nuclear YAP staining in wild-type lungs (*Figure 1S*). Consistent with the distribution of active YAP along the entire lung epithelium, expression of connective tissue growth factor (CTGF), an established YAP target, was detected in both the proximal and distal airways (*Figure 1T–V*). We did not observe a sharp transition of YAP from the cytoplasm to the nucleus at the junction between the SOX2$^+$ and SOX9$^+$ populations, a region dubbed the transition zone (TZ) (*Figure 1W*), as previously reported (*Mahoney et al., 2014*). In that study, inactive cytoplasmic YAP was detected in SOX2-expressing cells except those that abut SOX9-expressing cells at 12.5 and 14.5 *dpc*. We found that phosphorylated YAP (pYAP) was present in both SOX2$^+$ and SOX9$^+$ epithelial cells. pYAP levels were, in general, higher in the proximal than distal epithelium, but pYAP levels varied significantly from cell to cell in both the proximal and distal airways (*Figure 1—figure supplement 2*). Nevertheless, in many cells, low levels of pYAP were associated with the presence of nuclear YAP (*Figure 1—figure supplement 2*). Lung epithelial cells that do not have nuclear YAP usually stain positive for cytoplasmic YAP. This is consistent with a model in which pYAP is sequestered and degraded in the cytoplasm, but it also indicates a dynamic shuttling and distribution of YAP along the entire airway epithelium (*Chen et al., 2015*).

### Global deletion of *Yap* in the mouse lung epithelium results in defective lung branching morphogenesis and neonatal lethality

As a first step toward a mechanistic understanding of how Hippo signaling controls lung growth, we conditionally inactivated *Yap* in the lung epithelium using Cre lines that direct broad epithelial expression. We utilized the *Shh$^{Cre}$* line (*Harfe et al., 2004*) to convert a conditional (floxed) allele of *Yap* (designated as *Yap$^f$*) (*Xin et al., 2011*) to a null allele. The lungs of *Yap$^{f/f}$; Shh$^{Cre/+}$* embryos (called *Yap* mutants hereafter) (*Figure 2A–H*) consisted of a few large, thin-layered cysts, which replaced normal lung tissue and eliminated lung function (*Figure 2C,G,D,H*). This is similar to findings in an earlier report (*Mahoney et al., 2014*).

Defective lung development was already apparent at 11.5 *dpc* in *Yap$^{f/f}$; Shh$^{Cre/+}$* lungs (*Figure 2A,B,E,F*) prior to stereotyped branching and cell differentiation. The mutant lung buds failed to produce five fully separated buds. In particular, the buds that will produce the future cranial and middle lobes remained largely connected (*Figure 2B,N*; *Figure 3F*). This is likely due to the inability to undergo proper branching at very early stages of lung development. While stereotyped branching was actively underway in the wild-type lungs, the limited branching in *Yap* mutants came to a halt around 13.5 *dpc* (*Figure 2I–R*). In the absence of *Yap*, unbranched lung buds enlarged during development and turned into a multi-cyst structure. It is important to point out that cystic lung defects in *Yap* mutants that affect all lobes can be observed at very early stages of lung development, preceding expression of definitive epithelial markers. This would be consistent with a model in which YAP regulates the behavior of all epithelial cells.

To examine the early branching pattern, we dissected lungs from control and *Yap$^{f/f}$; Shh$^{Cre/+}$; ROSA26$^{mTmG/+}$* mice at 12.5 *dpc* and performed time-lapse microscopy on lungs grown at the air-liquid interface. At this stage, limited lung branching in *Yap*-deficient mutants would soon stop. While 'evaginations' (*Figure 2—figure supplement 1*; *Figure 2O,Q*) from existing lung buds were

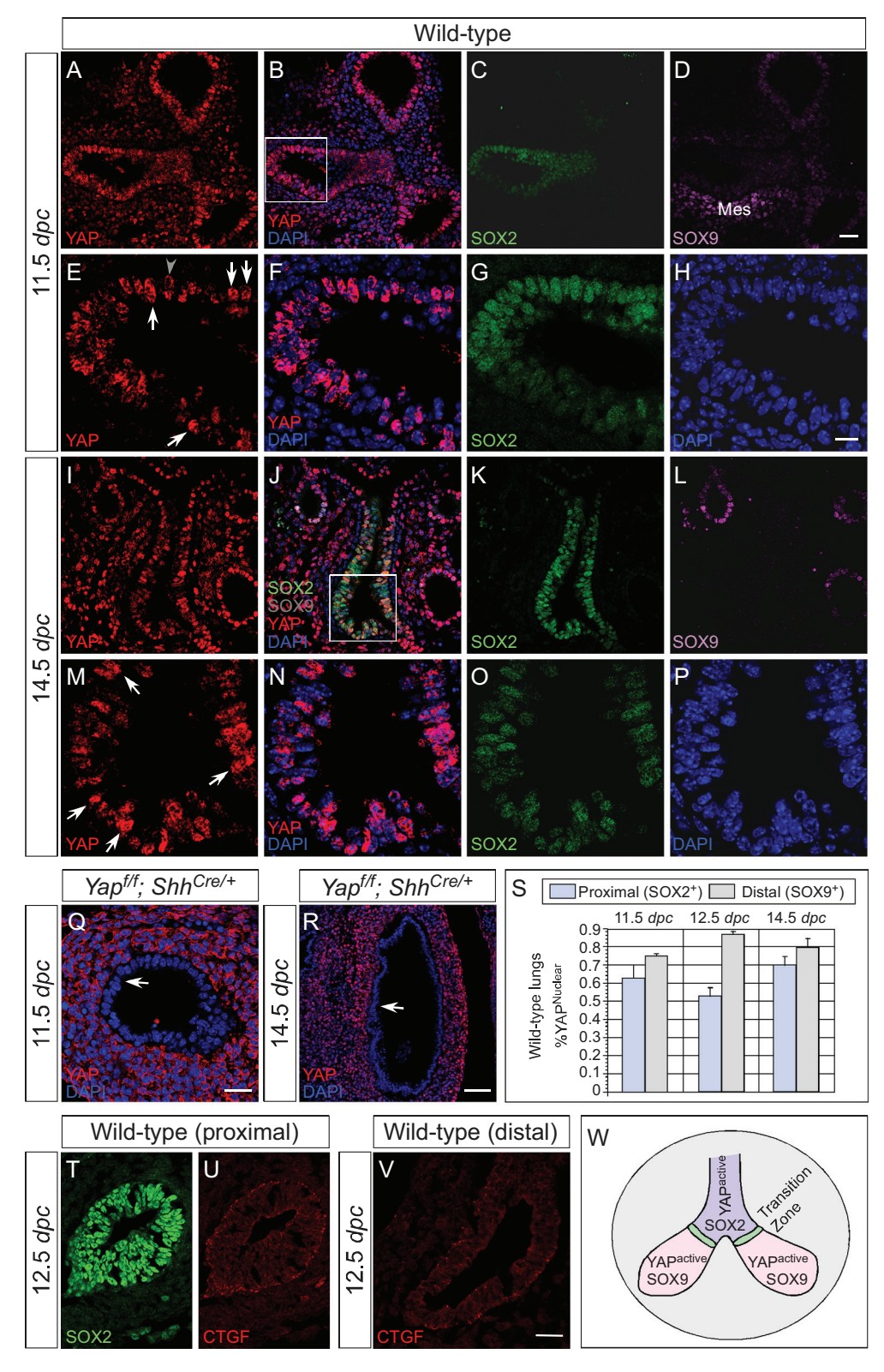

**Figure 1.** Nuclear YAP is active throughout the mouse lung epithelium during development. (**A–R**) Immunostaining of lung sections collected from wild-type and *Yap^{f/f}; Shh^{Cre/+}* mice at 11.5 and 14.5 *days post coitus* (*dpc*). The proximal airway is marked by SOX2 expression, while the distal airway is distinguished by SOX9 expression. Note that the strong SOX9 signal in the lower left corner of (**D**) is derived from the mesenchyme (mes). Nuclear YAP could be frequently found in both SOX2⁺ and SOX9⁺ domains and was not restricted to the 'transition zone', a small SOX2⁺ domain abutting the

*Figure 1 continued on next page*

*Figure 1 continued*

SOX9[+] compartment. The boxed regions in (B) and (J) were analyzed at a higher magnification as shown in (E–H) and (M–P), respectively. Representative epithelial cells with nuclear YAP (white arrows) or cytoplasmic YAP (grey arrowhead) are indicated in (E) and (M). Note that SOX2 and SOX9 staining is nuclear. YAP immunoreactivity was completely absent in the epithelium (arrow) but retained wild-type levels in the mesenchyme of $Yap^{f/f}$; $Shh^{Cre/+}$ mice (Q,R), demonstrating the specificity of YAP antibodies used in this study. (S) Quantification of lung epithelial cells with nuclear YAP in both the proximal and distal airways. A high percentage of cells exhibited nuclear YAP expression along the entire lung epithelium. A small fraction of epithelial cells with nuclear YAP also had cytoplasmic YAP. n = 8 for 11.5 *dpc*; n = 10 for 12.5 *dpc*; n = 10 for 14.5 *dpc*. (T–V) Immunostaining of lung sections collected from wild-type mice at 12.5 *dpc*. Expression of CTGF, a YAP target, was detected in both the proximal and distal airways. CTGF signal was barely detectable in $Yap^{f/f}$; $Shh^{Cre/+}$ lungs (not shown). (W) Schematic diagram that illustrates the distribution of active nuclear YAP throughout the entire lung epithelium. Scale bar = 25 μm for A–D, I–L; 10 μm for E–H, M–P; 25 μm for Q; 75 μm for R; 25 μm for T–V.

The following figure supplements are available for figure 1:

**Figure supplement 1.** Active nuclear YAP is distributed throughout the mouse lung epithelium during development.

**Figure supplement 2.** YAP and phospho-YAP are detected in both the proximal and distal airways during lung development.

noted in the absence of YAP, no new lungs bud were generated. Branching occurred in control lungs, but the rate of lung development was slowed down in *ex vivo* lung explants.

*Sox2* expression is greatly reduced in *Yap* mutant lungs. However, *Sox2*–deficient lungs do not display cystic phenotypes (*Que et al., 2009*; *Tompkins et al., 2009*), suggesting that lung cell-type specification and branching morphogenesis could be controlled by distinct sets of genes or pathways. Interestingly, *Shh* expression was reduced while *Fgf10* expression was upregulated in *Yap* mutant lungs (*Figure 2—figure supplement 2*). This is consistent with a negative regulatory relationship between *Shh* and *Fgf10* (*Pepicelli et al., 1998*), which mediate important interactions between lung epithelium and mesenchyme. How YAP controls *Shh* and *Fgf10* expression and interacts with major signaling pathways requires further investigation.

## Selective loss of YAP in restricted regions of the lung epithelium disrupts lung branching morphogenesis locally

The finding of nuclear localization of YAP throughout the lung epithelium led to our proposal that any given region of the lung epithelium is susceptible to loss of YAP and could produce local branching defects. In this model, the cystic lung phenotypes observed in $Yap^{f/f}$; $Shh^{Cre/+}$ mice originate from faulty morphogenesis along the entire lung epithelium due to global deletion of YAP (*Figure 1W*).

We reason that Cre lines that show more restricted or weaker epithelial expression than $Shh^{Cre}$ would allow us to test our model. To this end, we employed three additional Cre lines ($Sox9^{Cre}$, $spc^{Cre}$ and $Nkx2.1^{Cre}$) to inactivate $Yap^f$. $Sox9^{Cre}$ (*Akiyama et al., 2005*) displays restricted expression along the lung epithelium and would eliminate YAP in the SOX9[+] distal lung epithelium but not in the SOX2[+] proximal epithelium or the 'transition zone' (a small SOX2+ domain abutting the SOX9[+] compartment). Indeed, loss of YAP was found primarily in the distal airway, which is associated with selective loss of CTGF, a direct YAP target, in the distal airway (*Figure 3—figure supplement 1*). $Nkx2.1^{Cre}$ (*Xu et al., 2008*) and $spc^{Cre}$ (*Okubo and Hogan, 2004*) are supposed to be broadly expressed in the lung epithelium as suggested by Cre reporter activity) (*Song et al., 2012*). However, we noticed from our previous work (*Lin et al., 2015*) that both $Nkx2.1^{Cre}$ and $spc^{Cre}$ are less efficient than $Shh^{Cre}$ in converting a conditional allele to a null allele. $Nkx2.1^{Cre}$ is most active in the upper lobes, while $spc^{Cre}$ is most active in the distal epithelium. Thus, $Nkx2.1^{Cre}$ and $spc^{Cre}$ are expected to primarily delete *Yap* in limited areas of the lung where Cre has strong activity (*Figure 3—figure supplements 2 and 3*). Indeed, YAP removal in lung epithelial cells is correlated with Cre expression in these cells (*Figure 3—figure supplement 2*).

We set up matings and collected $Yap^{f/f}$; $Sox9^{Cre/+}$, $Yap^{f/f}$; $spc^{Cre/+}$ and $Yap^{f/f}$; $Nkx2.1^{Cre/+}$ lungs from 10.5 to 18.5 *dpc*. Phenotypic analysis was performed in a manner similar to that for $Yap^{f/f}$; $Shh^{Cre/+}$ lungs as described above. Our model predicts that disruption of YAP activity in a defined epithelial population will generate phenotypic consequences selectively in regions where YAP is lost. For instance, we predict that lung cysts will merely form in the distal part of $Yap^{f/f}$; $Sox9^{Cre/+}$ and

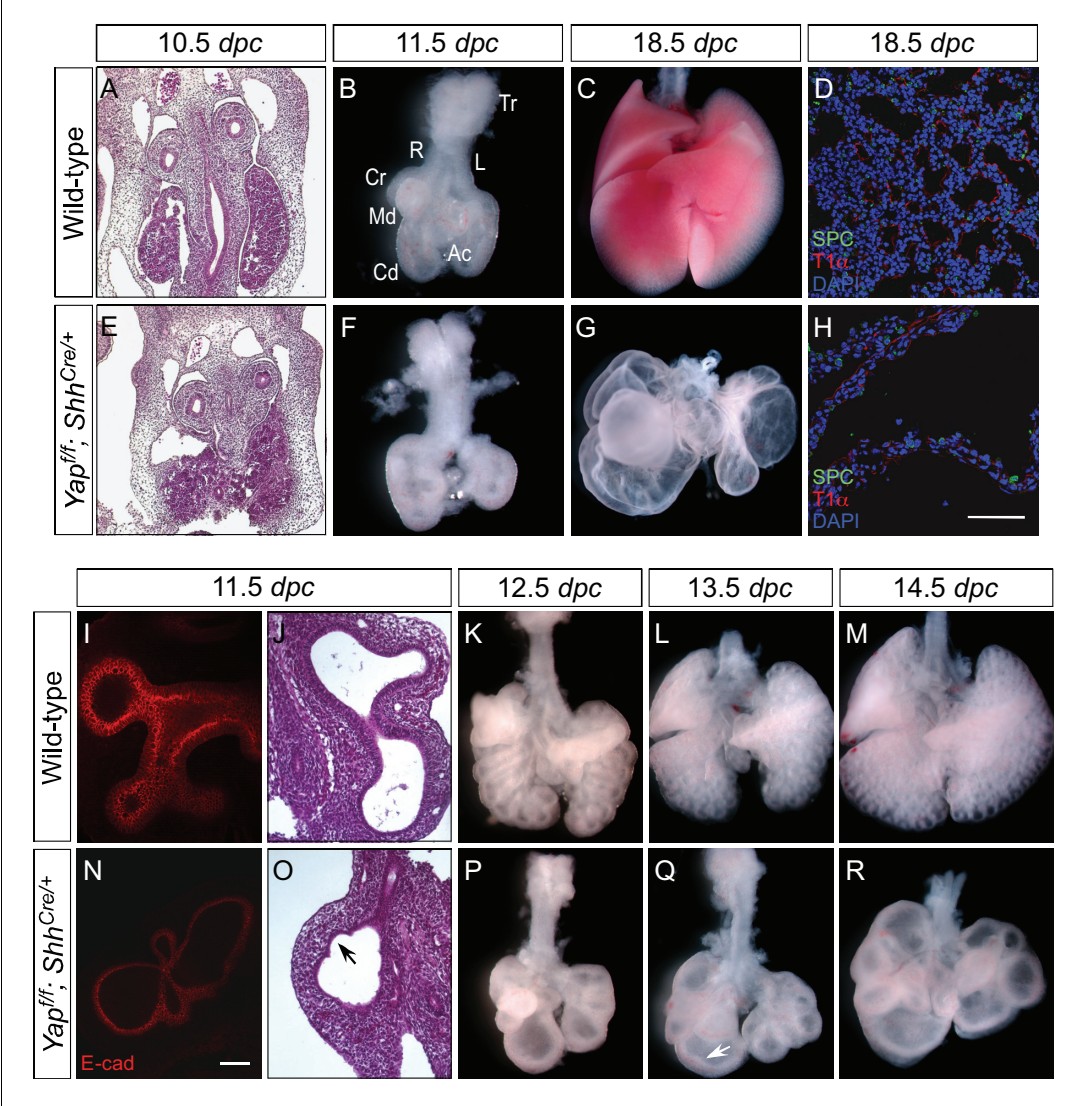

**Figure 2.** Loss of epithelial *Yap* leads to lung cysts. (A,E) Hematoxylin and eosin-stained sections of wild-type and *Yap*[f/f]; *Shh*[Cre/+] embryos at 10.5 *dpc*. No apparent difference in the morphology of lung buds was observed between wild-type and *Yap* mutants. (B,C,F,G) Ventral view of dissected lungs from wild-type and *Yap*[f/f]; *Shh*[Cre/+] mice at 11.5 and 18.5 *dpc*. Five buds were produced in both control and *Yap*-deficient lungs at 11.5 *dpc*; defective lung branching along the entire lung epithelium could already be detected at this stage in *Yap* mutants. As lung development proceeded, failure to execute a stereotyped program of branching in the absence of *Yap* resulted in lungs consisting only of multiple cysts at 18.5 *dpc*. R, right; L, left; Tr, trachea; Cr, cranial; Md. middle; Cd, caudal; Ac, accessory. (D,H) Immunostaining of lung sections collected from wild-type and *Yap*[f/f]; *Shh*[Cre/+] mice at 18.5 *dpc*. Cell types in the proximal airway of *Yap*[f/f]; *Shh*[Cre/+] mice failed to be specified. For instance, expression of markers for Clara [club] cells (CC10[+]), ciliated cells (acetylated-tubulin [Ac-tub][+]) and pulmonary neuroendocrine cells (CGRP[+]) were barely detectable (not shown). Reduction in the expression of distal lung cell markers, such as SPC (type II cells) and T1α (type I cells), in the cysts of *Yap*[f/f]; *Shh*[Cre/+] lungs was also noted. (I,N) Whole-mount immunostaining of wild-type and *Yap*[f/f]; *Shh*[Cre/+] lungs at 11.5 *dpc* by two-photon microscopy. Lung epithelium was identified by E-cadherin (E-cad). (J,O) Hematoxylin and eosin-stained sections of wild-type and *Yap*[f/f]; *Shh*[Cre/+] embryos at 11.5 *dpc*. The arrow points to 'evagination' of epithelial cells in *Yap*-deficient lung buds. (K–M, P–Q) Ventral view of dissected lungs from wild-type and *Yap*[f/f]; *Shh*[Cre/+] embryos at the developmental stages indicated. Epithelial 'evagination' (arrow in Q) could be seen in *Yap*-deficient lung buds but they failed to produce new buds subsequently. All views are ventral. Scale bar = 200 μm for C,G; 50 μm for D,H; 100 μm for I,N.

The following figure supplements are available for figure 2:

**Figure supplement 1.** Aborted branching in the absence of epithelial *Yap* in the lung.

**Figure supplement 2.** Changes in major signaling pathways in the absence of YAP.

*Figure 2 continued on next page*

*Figure 2 continued*

**Figure supplement 3.** Cell junctions and cell polarity are not disrupted due to the loss of epithelial *Yap* in the lung.

*Yap^{f/f}; spc^{Cre/+}* lungs, where Cre is capable of deleting *Yap* (***Figure 3—figure supplements 1*** and ***2***). Similarly, lung cysts will be generated in the upper lobes of *Yap^{f/f}; Nkx2.1^{Cre/+}* lungs (***Figure 3— figure supplement 3***).

We found that *Yap^{f/f}; Sox9^{Cre/+}*, *Yap^{f/f}; spc^{Cre/+}* and *Yap^{f/f}; Nkx2.1^{Cre/+}* lungs at 18.5 *dpc* all contained cysts of varying sizes (***Figure 3A–D***; ***Figure 3—figure supplement 4***). Importantly, lung cysts (arrows) only formed in the distal part of *Yap^{f/f}; Sox9^{Cre/+}* and *Yap^{f/f}; spc^{Cre/+}* lungs while lung cysts (arrows) were observed in the upper lobes of *Yap^{f/f}; Nkx2.1^{Cre/+}* lungs (***Figure 3A–D***; ***Figure 3—figure supplement 4***). Similar conclusions were reached when branching defects and cyst formation were examined at earlier stages of lung development (***Figure 3E–N***; ***Figure 3—figure supplement 4***). For instance, defective branching was apparent in *Yap^{f/f}; Shh^{Cre/+}* lungs at 11.5 *dpc*. Branching defects and cyst formation in the distal airways could be discerned by 12.5 *dpc* in *Yap^{f/f}; spc^{Cre/+}* lungs, while branching defects and cyst formation in the distal airways of *Yap^{f/f}; Sox9^{Cre/+}* lungs did not appear until 13.5 *dpc*. Of note, the cystic lung phenotype in *Yap^{f/f}; Nkx2.1^{Cre/+}* mice was not apparent until 14.5 *dpc*.

Not surprisingly, two copies of *Nkx2.1^{Cre}* enhanced the cystic lung phenotype, consistent with low levels of *Nkx2.1^{Cre}* activity in converting *Yap^f* into a null allele (***Figure 3—figure supplement 5***). In fact, two copies of *spc^{Cre}* also enhanced the cystic lung phenotype, indicating a failure of *Yap* removal by *spc^{Cre}* in many epithelial cells. Together, these findings indicate that lung cysts can develop at a restricted area along the lung epithelium. They are consistent with our model in which YAP controls local epithelial cell properties (***Figure 1W***) and loss of YAP in a given location of the lung epithelium can disrupt lung development, leading to local cyst formation.

## Local cyst formation in the absence of YAP is independent of proximal cell-type specification

Our model suggests that cyst formation in the absence of *Yap* is due to changes in local cell properties. We expect that cell-type specification would only be affected in regions where defective branching is caused by loss of YAP. To test this idea, we took advantage of *Yap^{f/f}; Sox9^{Cre/+}* and *Yap^{f/f}; spc^{Cre/+}* lungs in which the lung cysts are located in the distal part of the lung. We performed marker analysis on *Yap^{f/f}; Sox9^{Cre/+}* and *Yap^{f/f}; spc^{Cre/+}* lungs collected at 10.5–18.5 *dpc* to determine lung cell-type specification. If YAP controls lung development by regulating local epithelial cell properties, we anticipate that disruption of local branching in the distal epithelium of *Yap^{f/f}; Sox9-^{Cre/+}* and *Yap^{f/f}; spc^{Cre/+}* lungs will not disturb specification of proximal lung cell types.

Indeed, we discovered that expression of SOX2 and markers of proximal cell types appeared to be well maintained in *Yap^{f/f}; Sox9^{Cre/+}* and *Yap^{f/f}; spc^{Cre/+}* (and even in *Yap^{f/f}; Nkx2.1^{Cre/+}*) lungs (***Figure 4A–H***; ***Figure 4—figure supplement 1***). For instance, production of lung cell types in the more proximal airways, such as Clara [club] cells (CC10 [Scgb1a1]^+), ciliated cells (Ac-tubulin^+) and pulmonary neuroendocrine cells (CGRP^+), in *Yap^{f/f}; Sox9^{Cre/+}* and *Yap^{f/f}; spc^{Cre/+}* lungs was indistinguishable from that in control lungs. As expected from distal cyst formation, the generation of distal cell types, including alveolar type II cells (SPC^+, SPB^+, SPD^+) and type I cells (T1α^+, Aqp5^+), was severely affected in *Yap^{f/f}; Sox9^{Cre/+}* and *Yap^{f/f}; spc^{Cre/+}* lungs. This suggests that disruption of branching is associated with defective cell-type specification locally. In addition, cyst formation can be independent of proximal cell-type specification.

## Cell junctions and cell polarity are not altered in *Yap*-deficient lungs

YAP is associated with cell junction proteins, and it was proposed that YAP activity is not only controlled by signals from the cell junction and the actin cytoskeleton but may also function in a feedback loop to control cell polarity and the cytoskeleton (***Low et al., 2014***; ***Boggiano and Fehon, 2012***; ***Schroeder and Halder, 2012***). Any defects in these essential cellular processes could underlie cystic lung development. Consistent with this notion, cysts form in *Cdc42*-deficient lungs in which apical-basal polarity is disrupted (***Wan et al., 2013***).

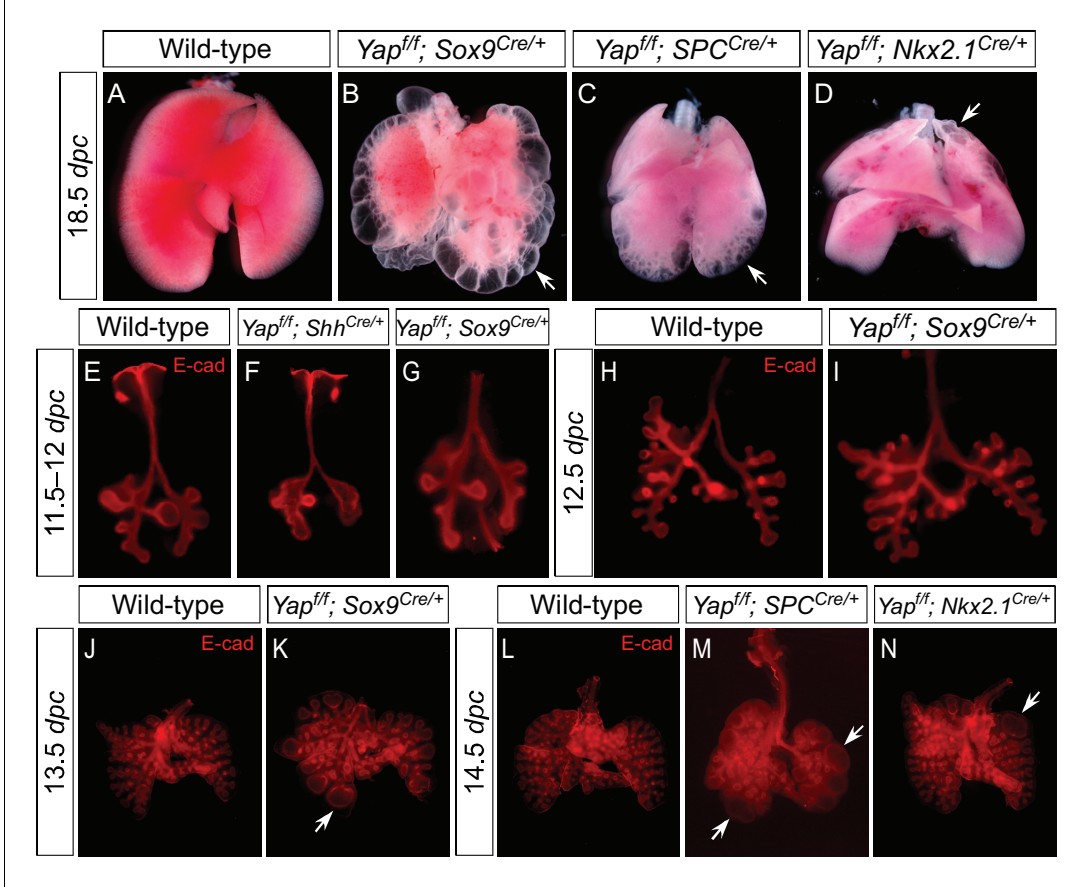

**Figure 3.** Regional loss of epithelial *Yap* leads to localized lung cysts. (**A–D**) Ventral view of dissected lungs from wild-type, *Yap^f/f; Sox9^Cre/+*, *Yap^f/f;* *spc^Cre/+* and *Yap^f/f; Nkx2.1^Cre/+* mice at 18.5 *dpc*. Lung cysts in *Yap^f/f; Sox9^Cre/+* and *Yap^f/f; spc^Cre/+* mice were largely confined to the distal airway (arrows in **B,C**), while lung cysts were found in the upper lobes (arrow in **D**) of *Yap^f/f; Nkx2.1^Cre/+* mice. The location of lung cyst formation is correlated with the sites of strong Cre activity and *Yap* removal. This suggests that loss of *Yap* at a given region leads to localized lung cysts. (**E–N**) Whole-mount immunostaining of dissected lungs from wild-type, *Yap^f/f; Shh^Cre/+*, *Yap^f/f; Sox9^Cre/+*, *Yap^f/f; spc^Cre/+* and *Yap^f/f; Nkx2.1^Cre/+* mice at the stages indicated. Lung epithelium was visualized by E-cadherin (E-cad). While defective branching was apparent in *Yap^f/f; Shh^Cre/+* lungs at 11.5 *dpc*, branching defects and cyst formation in *Yap^f/f; Sox9^Cre/+* lungs did not appear until 13.5 *dpc*. Cyst formation in *Yap^f/f; Sox9^Cre/+* lungs was confined to the distal airways (arrow in **K**). Similarly, cyst formation in *Yap^f/f; spc^Cre/+* lungs was detected primarily in the distal airways (arrows in **M**). By contrast, cyst formation was found in the upper lobes (arrow in **N**) of *Yap^f/f; Nkx2.1^Cre/+* lungs. This suggests that loss of *Yap* at a given region leads to localized lung cysts. All views are ventral.

The following figure supplements are available for figure 3:

**Figure supplement 1.** Deletion of *Yap* in the distal lung epithelium by the *Sox9^Cre* mouse line.

**Figure supplement 2.** Expression of *spc^Cre* is associated with loss of YAP in SOX9⁺ distal airways.

**Figure supplement 3.** Expression of *Nkx2.1^Cre* is associated with loss of YAP in the upper lobes.

**Figure supplement 4.** Loss of epithelial *Yap* at restricted areas leads to localized lung cysts.

**Figure supplement 5.** The dosage of *Nkx2.1^Cre* and *spc^Cre* affects the severity of lung phenotypes.

To test whether YAP regulates cell polarity, we investigated the distribution of aPKC (an apical marker), ZO1/ZO2 (subapical markers) and Laminin (a basolateral marker) in control and *Yap* mutant lungs. We showed that lung branches or cysts with proper apical aPKC, subapical ZO1/ZO2 and basolateral Laminin could be observed in *Yap* mutants at 11.5, 12.5 and 14.5 *dpc* (**Figure 2—figure**

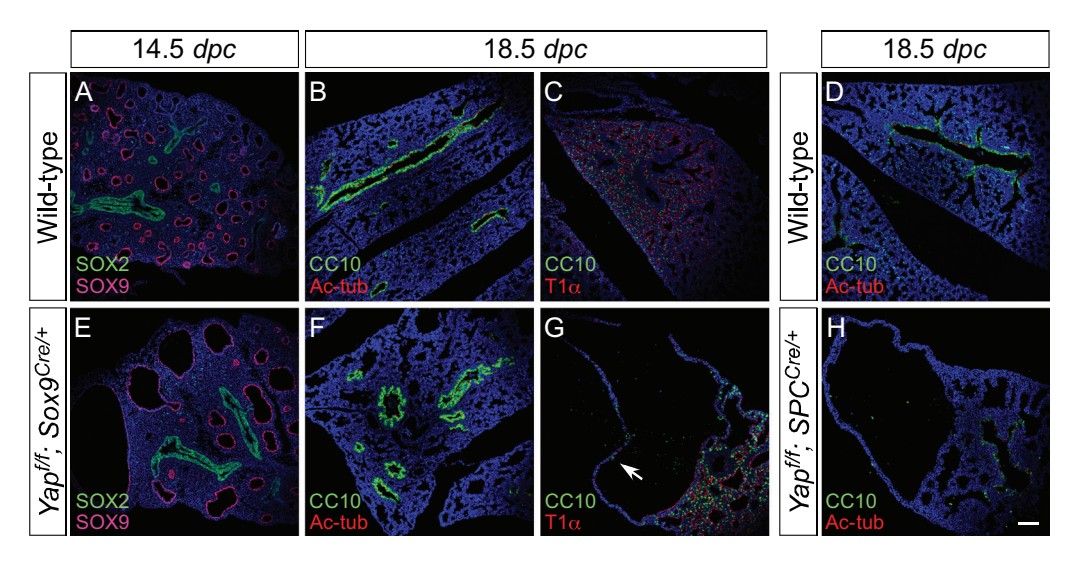

**Figure 4.** Proximal airway development is unaffected by loss of YAP in the distal airway. (A–H) Immunostaining of lung sections collected from wild-type, $Yap^{f/f}$; $Sox9^{Cre/+}$ and $Yap^{f/f}$; $spc^{Cre/+}$ mice at 14.5 and 18.5 $dpc$ as indicated. Proximal-distal airway specification was unaffected in $Yap^{f/f}$; $Sox9^{Cre/+}$ lungs as revealed by proper expression of the proximal (SOX2) and distal (SOX9) airway markers despite cyst formation in the SOX9$^{+}$ lung epithelium. Moreover, cell types in the proximal airway of $Yap^{f/f}$; $Sox9^{Cre/+}$ and $Yap^{f/f}$; $spc^{Cre/+}$ mice were specified. For example, cells expressing CC10 (Clara cell marker) and acetylated-tubulin [Ac-tub] (ciliated cell marker) were detected in a similar pattern between control and mutant lungs. Scale bar = 100 μm for A–H.

The following figure supplement is available for figure 4:

**Figure supplement 1.** Disruption of the distal airway in $Yap^{f/f}$; $spc^{Cre/+}$ and $Yap^{f/f}$; $Sox9^{Cre/+}$ mouse lungs.

*supplement 3*). This suggests that lung epithelial cell polarity is not disrupted in the absence of YAP.

We also examined possible changes in cell junctions in *Yap*-deficient lungs. No apparent alterations in markers for tight junctions (e.g. ZO1, ZO2) and adherens junctions (e.g. E-cadherin, β-catenin and α-catenin) were found between control and *Yap* mutant lungs (*Figure 2—figure supplement 3*). Thus, loss of YAP does not perturb cell junctions at the light microscopy level. These results suggest that other cellular mechanisms are responsible for cyst formation in the absence of YAP.

## RNA-Seq and ChIP-Seq analysis of embryonic lungs uncovers new YAP-responsive genes that are involved in regulating the cell cycle and cellular contractility

To understand how disruption of local epithelial properties leads to defective lung branching and cyst formation, we performed RNA-Seq analysis on control and *Yap*-deficient lungs at 12.5 and 14.5 *dpc*, and similar results were obtained at both stages. Genes that control the cell cycle and cellular contractility topped the cluster of differentially expressed genes (*Figure 5A,C*). Genes described in this study were verified by qPCR analysis (*Figure 5—figure supplement 1*).

YAP is a transcriptional coactivator without intrinsic DNA-binding activity and YAP functions in conjunction with transcription factors TEAD1–4 to activate Hippo pathway targets. To systematically search for YAP targets during lung development, we performed ChIP-Seq analysis on wild-type lungs at 14.5 *dpc*. We showed that known YAP targets (such as *Ctgf*, *Ajuba* and *Amotl2*) were identified in this approach (*Figure 5B*; *Figure 5—figure supplement 2*). Many new YAP targets were also uncovered.

Analysis of both RNA-Seq and ChIP-Seq data allowed us to identify a list of YAP targets that not only have a reduced expression in *Yap* mutant lungs but also have a TEAD-binding site within the

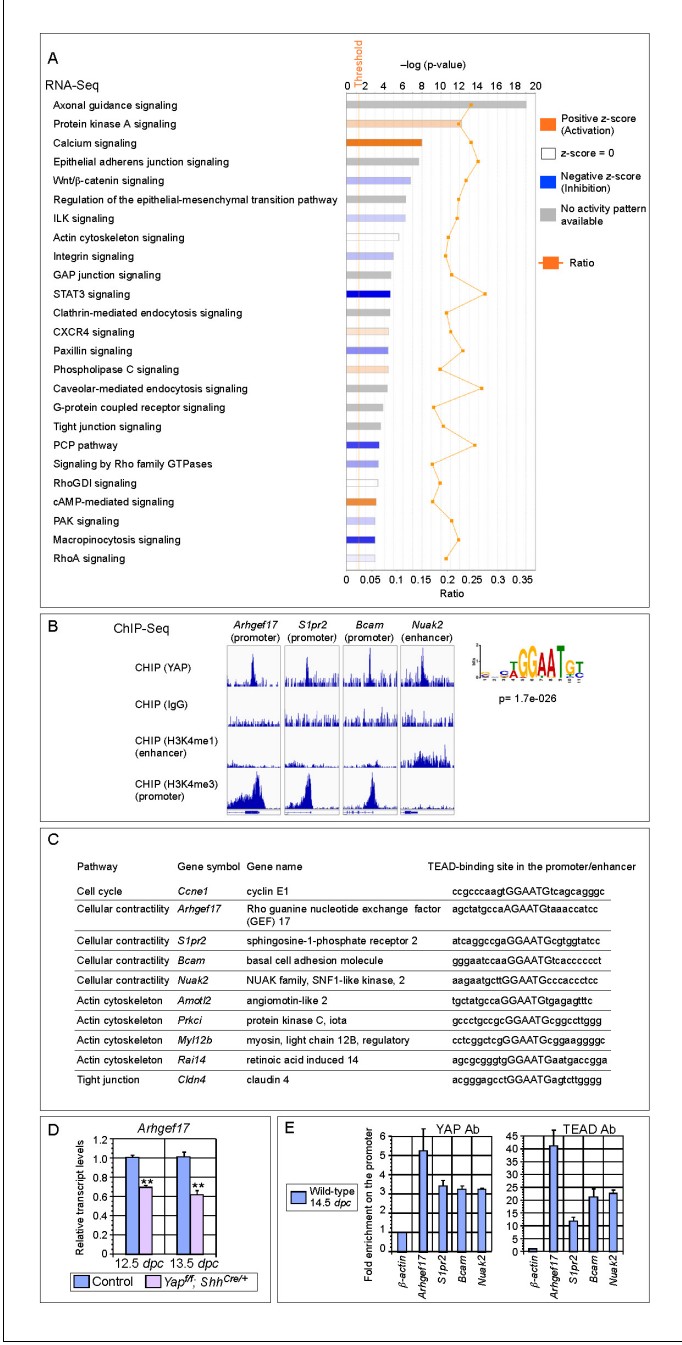

**Figure 5.** Genome-wide expression analysis identifies YAP targets that are involved in regulating cell proliferation and cellular contractility. (**A**) Pathway analysis of differentially expressed genes identified in RNA-Seq analysis of control and *Yap*-deficient lungs at 12.5 *dpc*. p-value and z-score were shown. The calculated z-score indicates the prediction of overall increase or decrease in the activity of a pathway. For z-score >0, the pathway is predicted to be activated; for z-score <0, the pathway is predicted to be inhibited. The ratio indicates the ratio of genes from the dataset that map to the pathway divided by the total number of genes that map to the same pathway. The orange line indicates a threshold of -log(p-value) = 1.30 (p<0.05) and the cutoff was set at -log(p-value) = 3 (p<0.001). Several pathways involved in regulating the cell cycle, cellular contractility and cell adhesion were uncovered. (**B**) Visualization of YAP-enriched peaks in ChIP-Seq for the indicated genes using Integrative Genomics Viewer (IGV). The peaks were associated with the promoter or enhancer as determined by ChIP-Seq for histone modifications. The conserved TEAD-binding site that is present in YAP-enriched peaks was also shown. (**C**) A list of *bona fide* YAP targets in the lung. These YAP targets not only contain a TEAD-binding site in their promoter/enhancer but their expression was also reduced in *Yap*-deficient lungs by RNA-Seq or qPCR analysis.

*Figure 5 continued on next page*

*Figure 5 continued*

Many of these YAP targets are involved in regulating cell proliferation and cellular contractility. (D) qPCR analysis of *Arhgef17* in wild-type and *Yap^{f/f}; Shh^{Cre/+}* lungs (n ≥ 3 for each group) at 12.5 and 13.5 *dpc*. The mRNA levels of *Arhgef17* were significantly reduced in the absence of *Yap*. Note that mRNA from the whole lung was used for qPCR analysis and *Arhgef17* in the mesenchyme was presumably unaffected by *Shh^{Cre}*. All values are means ± SEM. (**) p<0.01 (unpaired Student's *t*-test). (E) ChIP-qPCR of YAP targets identified from ChIP-Seq of wild-type lungs. Both YAP and TEAD were found to reside at the promoter of YAP targets. We also found that neither YAP nor TEAD was enriched on the *Sox2* promoter (not shown), suggesting that YAP does not directly regulate *Sox2* expression in the developing lung.

The following figure supplements are available for figure 5:

**Figure supplement 1.** Analysis of the transcript levels of YAP targets in the mouse lung.
**Figure supplement 2.** ChIP analysis identifies new YAP target genes in the mouse lung.

---

ChIP-Seq peak in their promoter/enhancer. Many of these YAP targets are involved in regulating the cell cycle (e.g. *Ccne1*) and cellular contractility (such as *Arhgef17*, *Bcam*, *S1pr2* and *Nuak2*) (*Figure 5C–E*). These genes are candidates to mediate YAP activity in controlling epithelial cell properties. While cell adhesion/tight junctions and the cytoskeleton have been shown to influence Hippo signaling (*Matsui and Lai, 2013*; *Dupont et al., 2011*), our new findings suggest that Hippo signaling reciprocally controls the actomyosin cytoskeleton and mechanical force production. This could transduce essential functions of YAP during lung development.

## The rate of cell proliferation is reduced while cell death is unaltered in the absence of *Yap*

To explore the molecular mechanisms that underlie the *Yap* mutant phenotypes, we investigated whether cell proliferation was affected in *Yap*-deficient lungs. To this end, we quantified BrdU+ or EdU+ cells in the epithelial and mesenchymal compartments of control and *Yap*-deficient lungs at 11.5, 12.5 and 14.5 *dpc*. This method provided a more accurate assessment of cell proliferation than using Ki67, PH3 or PCNA since only cells in the S phase will be selectively labeled. We noticed that the rate of cell proliferation in *Yap*-deficient lung epithelium was reduced, while mesenchymal cell proliferation was unaffected in the absence of YAP (*Figure 6A–C* and data not shown). Decreased cell proliferation in *Yap* mutant lungs would lower the number of epithelial cells and contribute to the failure of lung branching in *Yap* mutants.

In contrast, the rate of cell death by TUNEL assay was unaltered in the absence of *Yap* (data not shown). Thus, cell death is neither a key factor in mediating YAP activity in the lung nor a contributor to cyst formation in the absence of YAP.

## Mechanical force production is impaired in *Yap* mutant lungs

RNA-Seq analysis of *Yap*-deficient lungs revealed control of cellular contractility by Hippo signaling. Moreover, actomyosin-mediated contraction was shown to regulate lung branching in chick embryos (*Kim et al., 2013*). To study whether YAP controls cellular contractility and mechanical force production, we determined the expression of pMLC (phosphorylated myosin light chain) in control and *Yap* mutant lungs. pMLC is indicative of contractility mediated by actomyosin. We found that pMLC protein levels were significantly diminished in *Yap^{f/f}; Shh^{Cre/+}* lungs (*Figure 7A*; *Figure 7—figure supplement 1*). This led to our speculation that cortical pMLC, which is associated with mechanical force production during morphogenesis is disrupted in the absence of *Yap*. To test this idea, we utilized whole-mount immunofluorescence and two-photon microscopy to visualize the distribution of cortical pMLC. In addition, we focused on control and *Yap^{f/f}; Shh^{Cre/+}* lungs at 11.5 *dpc* when the branching defects were about to take place (*Figure 7B,C*). This would allow us to identify the driving force of defective branching. In wild-type lungs, cortical pMLC was distributed in both epithelial and mesenchymal cells at 11.5 *dpc* (*Figure 7D–F*). By contrast, cortical pMLC was drastically reduced in the epithelium but retained normal levels in the mesenchyme of *Yap^{f/f}; Shh^{Cre/+}* lungs at 11.5 *dpc* (*Figure 7G–I*).

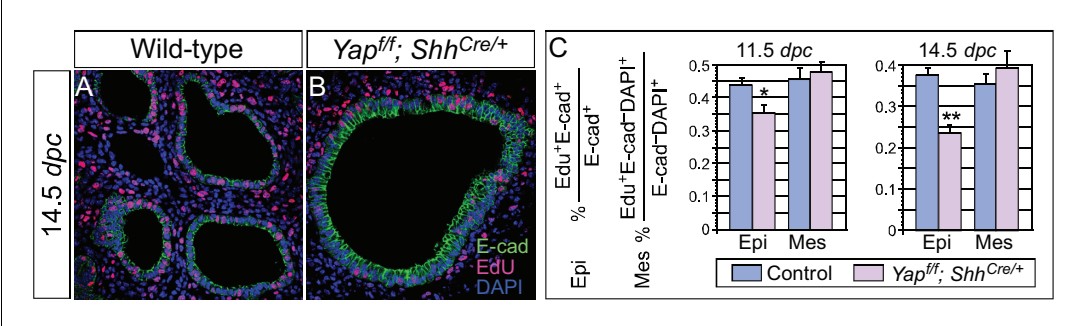

**Figure 6.** Cell proliferation is reduced in the epithelium of *Yap*-deficient lungs. (**A,B**) Immunostaining of lung sections collected from wild-type and *Yap^{f/f}; Shh^{Cre/+}* mice injected with EdU at 14.5 *dpc*. Lung epithelial cells were distinguished by E-cadherin (E-cad) staining. (**C**) Quantification of cell proliferation rate in the epithelium (Epi) and mesenchyme (Mes) of control and *Yap^{f/f}; Shh^{Cre/+}* lungs at 11.5 and 14.5 *dpc*. The rate of epithelial cell proliferation was calculated as the ratio of the number of EdU⁺ epithelial cells (EdU⁺E-cad⁺) to the number of epithelial cells (E-cad⁺). An apparent reduction in the percentage of EdU⁺ cells was detected in the epithelium of *Yap^{f/f}; Shh^{Cre/+}* lungs compared to controls (n ≥ 8 for each group). By contrast, cell proliferation in the mesenchyme, where YAP was untouched by *Shh^{Cre}*, was indistinguishable between control and *Yap^{f/f}; Shh^{Cre/+}* lungs. The rate of mesenchymal cell proliferation was calculated as the ratio of the number of EdU⁺ mesenchymal cells (EdU⁺E-cad⁻ DAPI⁺) to the number of mesenchymal cells (E-cad⁻ DAPI⁺). All values are means ± SEM. (*) p<0.05; (**) p<0.01 (unpaired Student's *t*-test). We found that most epithelial cells in control or *Yap*-deficient lungs at 11.5 *dpc* expressed Ki67. This is likely due to a short cell cycle at this stage, which makes Ki67 (as well as other commonly used markers) unsuitable for accurate detection of differences in cell proliferation.

To further test the role of YAP in mechanical force production during lung branching, we measured mechanical force generation in the absence of YAP. We performed laser ablation of a strip of GFP-labeled epithelial cells in control and *Yap^{f/f}; Shh^{Cre/+}; ROSA26mTmG* lungs using two-photon microscopy (**Figure 7J–M**). Epithelial cell loss triggered recoil of the surrounding cells from the ablation site in control lungs (**Figure 7N,O**; **Video 1**), which was followed by gradual movement of neighboring cells to cover the gap. By contrast, no tissue recoil was observed in *Yap*-deficient lungs following laser ablation (**Figure 7P,Q**; **Video 2**). The initial velocity of the recoil in the first few seconds post-ablation is shown to be proportional to the magnitude of the resting tension (**Hutson et al., 2003**). Thus, measuring the recoil of the surrounding cells within the first few seconds of laser ablation would allow us to probe the mechanical state of the lung. We found that the velocity of recoil was reduced in *Yap*-deficient lungs compared to that of control lungs (**Figure 7R**). This is consistent with compromised mechanical force production in the absence of YAP.

We propose that reduction in cortical pMLC at very early stages of lung development (prior to cell differentiation) would compromise mechanical force production and lead to defective branching. This further suggests that altered epithelial cell properties underlie the lung abnormalities in *Yap* mutants.

## YAP regulates pMLC levels and mechanical force production via multiple pathways

Control of actin and pMLC is mediated by the RhoA–ROCK cascade (**Amano et al., 2010**) (**Figure 7S**). The small GTPase Rho binds and activates ROCK (Rho-associated protein kinase). ROCK is the major regulator of the actomyosin cytoskeleton and ROCK enhances myosin phosphorylation and consequently cellular contractility. RhoA activity is regulated by RhoGEF (Rho guanine exchange factor), which activates RhoA, and RhoGAP (Rho GTP activating factor), which inhibits RhoA.

To investigate the molecular mechanisms by which YAP regulates pMLC levels, we searched for RhoGEFs that are downregulated in *Yap* mutant lungs. We found that the mRNA levels of *Arhgef17* (encoding a RhoGEF) were reduced in *Yap*-deficient lungs by RNA-Seq and qPCR analysis (**Figure 5D**). Notably, ChIP-Seq analysis of YAP identified a peak in the *Arhgef17* promoter, which contains a conserved TEAD-binding site. We showed that both YAP and TEAD reside in the *Arhgef17* promoter in embryonic lungs at 14.5 *dpc* by ChIP-qPCR analysis (**Figure 5E**), indicating that *Arhgef17* is a direct YAP target. This suggests that ARHGEF17 is an effector of YAP in controlling a Rho–ROCK-pMLC cascade and mechanical force production.

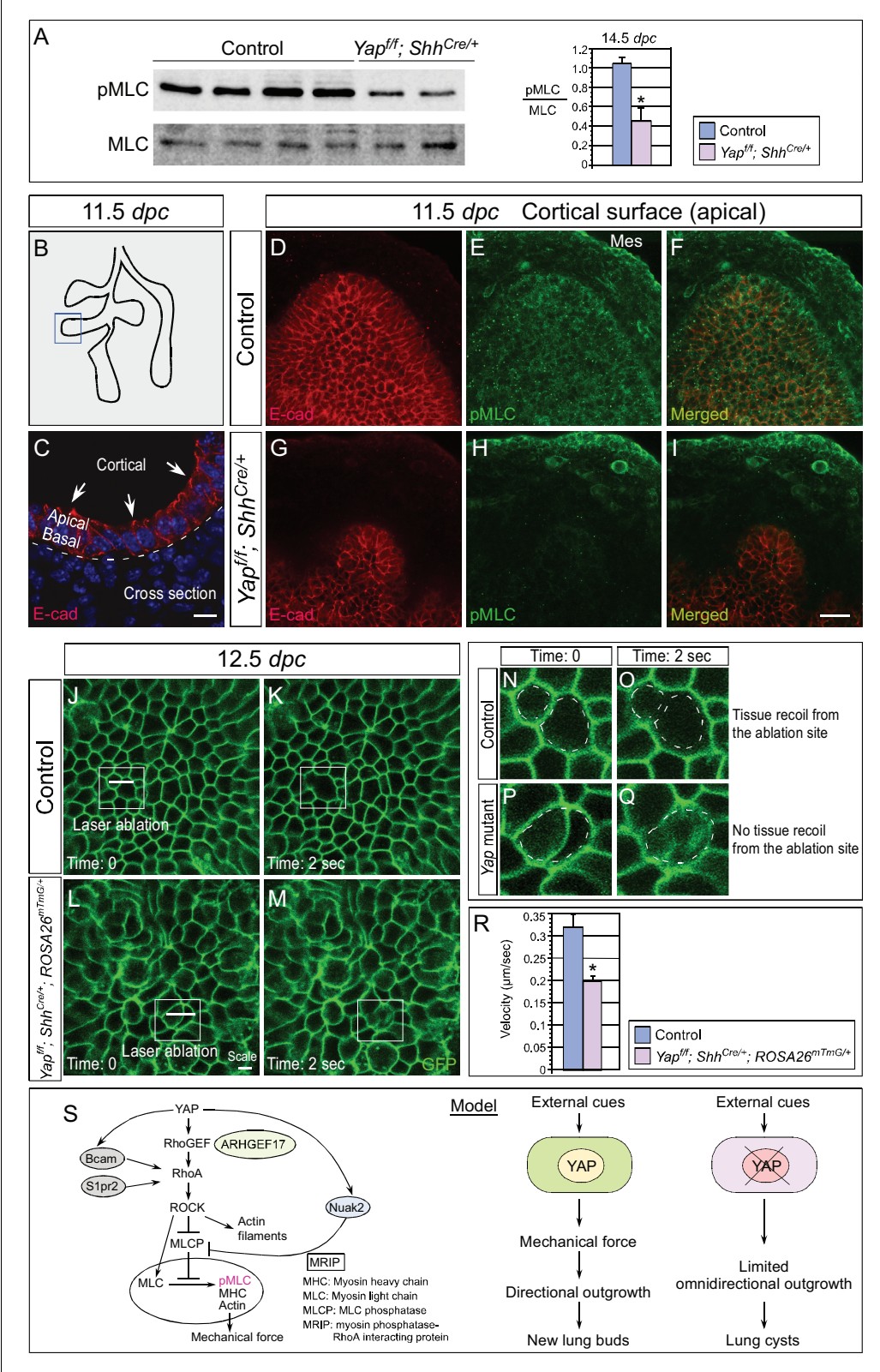

**Figure 7.** Cortical pMLC is greatly reduced in the epithelium of *Yap*-deficient lungs and mechanical force production is compromised. (**A**) Western blots of lysates derived from control and *Yap^{f/f}; Shh^{Cre/+}* lungs at 14.5 *dpc*. Protein levels of phosphorylated myosin light chain (pMLC) were significantly reduced in the absence of *Yap*. Moreover, the ratio of pMLC to MLC protein levels was also diminished in *Yap* mutant lungs. (**B**) Schematic diagram of wild-type lungs at 11.5 *dpc*. The boxed region indicates areas shown in **D–I**. (**C**) Cross-section of a lung bud that illustrates the apical and basal surface

*Figure 7 continued on next page*

*Figure 7 continued*

and cortical view along the apical surface. (**D–I**) Whole-mount immunostaining of control and *Yap^{f/f}; Shh^{Cre/+}* lungs at 11.5 *dpc* by two-photon microscopy. This enabled visualization of the distribution of pMLC along the cortical surface of epithelial cells located at the apical surface of the lung bud. Lung epithelium was identified by E-cadherin (E-cad). Cortical pMLC was detected in both the epithelium and mesenchyme (mes) of control lungs. By contrast, cortical pMLC could not be detected in the epithelium but retained wild-type levels in the mesenchyme of *Yap^{f/f}; Shh^{Cre/+}* lungs. (**J–M**) Laser ablation of control and *Yap^{f/f}; Shh^{Cre/+}; ROSA26^{mTmG/+}* lungs. Lung epithelial cells were labeled by GFP from the *ROSA26^{mTmG/+}* allele induced by *Shh^{Cre}*. Laser ablation of lung epithelial cells by two-photon microscopy (indicated by bars in **J,L**) triggered recoiling of non-injured neighboring cells in control but not in *Yap*-deficient lungs (see *Videos 1* and *2*). Snapshots of the lung epithelium post-ablation were shown. The boxed regions in (**J–M**) indicate areas shown in (**N–Q**). (**N–Q**) The dotted lines in (**N,O**) demarcate the cell boundary prior to laser ablation in control lungs. Surrounding cells recoiled from the ablation site as indicated by the space between the dotted line and the new position of cells (**O**). By contrast, no tissue recoil was found in *Yap* mutant lungs (**Q**). (**R**) The velocity of epithelial recoil was measured in control and *Yap*-deficient lungs after laser ablation. A reduction in velocity was found in the absence of YAP (n ≥ 6 for each group). All values are means ± SEM. (*) p<0.05 (unpaired Student's *t*-test). (**S**) Schematic diagram of signaling cascades that control the production of pMLC and mechanical force. Our model suggests that YAP regulates multiple pathways to regulate pMLC generation. In addition to a YAP-ARHGEF17-RhoA-ROCK pathway, YAP also induces the expression of *Bcam*, *S1pr2* and *Nuak2* to enhance pMLC levels. BCAM and S1PR2 activate RhoA, while NUAK2 inhibits MLCP (myosin light chain phosphatase) in a RhoA-independent manner. The employment of a signaling network ensures the production of appropriate amounts of pMLC required for lung branching. Mechanical force production is perturbed in *Yap*-deficient lungs due to reduced pMLC. This would contribute to defective lung branching and cyst formation. Scale bar = 10 µm for **C**; 25 µm for **D–I**.

The following figure supplement is available for figure 7:

**Figure supplement 1.** Reduction in pMLC protein levels in *Yap* mutant lungs.

Dynamic lung branching requires intricate control of cellular contractility. We surmise that YAP controls pMLC levels and mechanical force production through multiple pathways, including a YAP–RhoGEF–RhoA–ROCK pathway (*Figure 7S*). Indeed, our RNA-Seq and ChIP-Seq analyses uncovered several new YAP targets that regulate pMLC levels via different mechanisms. They include the *Laminin alpha 5 receptor* (*Bcam*), *Sphingosine-1-phosphate receptor 2* (*S1pr2*) and *NUAK family, SNF1-like kinase, 2* (*Nuak2*). Expression of these genes was downregulated in the absence of YAP. Furthermore, conserved TEAD-binding sites were found in the promoters of *Bcam*, *S1pr2* and *Nuak2* (*Figure 5C*), where the YAP and TEAD proteins reside as revealed by ChIP-Seq and ChIP-qPCR analysis (*Figure 5E*). BCAM and S1PR2 can activate RhoA (*Collec et al., 2011*; *Randriamboavonjy et al., 2009*) to promote pMLC production. The NUAK2 kinase can also enhance pMLC generation by inhibiting MLC phosphatase independently of RhoA (*Zagórska et al., 2010*). Our results suggest that YAP could control pMLC levels and mechanical force production through an elaborate signaling network (*Figure 7S*).

## Discussion

Our work provides new insight into the molecular mechanisms by which YAP controls lung development. We propose that reduced cell number and disrupted mechanical force generation contribute to defective branching in *Yap* mutant lungs. Previous studies reported abnormal lung branching through pharmacological manipulation of Rho, ROCK or the actomyosin cytoskeleton in *ex vivo* lung culture (*Moore et al., 2005*). This is consistent with our

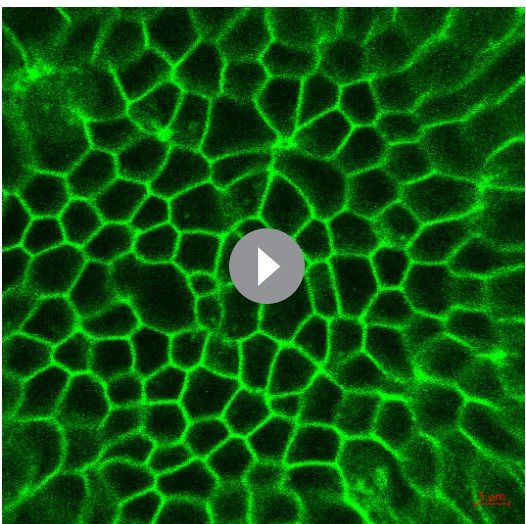

**Video 1.** Movement of surrounding cells after laser ablation of lung epithelial cells in control mice. Immediately following laser ablation by two-photon microscopy, movement of epithelial cells (labeled by GFP) surrounding the injured cells were imaged at one frame per sec (fps) for 50 s. The movie was played back at 10 pfs. *Video 1* is related to *Figure 7J,K*.

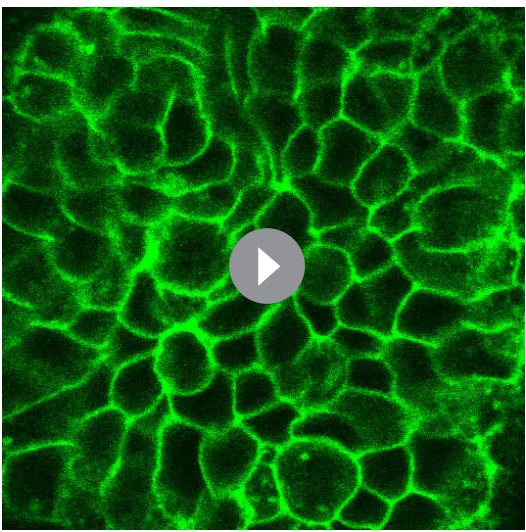

**Video 2.** Movement of surrounding cells after laser ablation of lung epithelial cells in *Yap^f/f^; Shh^Cre/+^; ROSA26^mTmG/+^* mice. Immediately following laser ablation by two-photon microscopy, movement of epithelial cells (labeled by GFP) surrounding the injured cells were imaged at one frame per sec (fps) for 50 s. The movie was played back at 10 pfs. *Video 1* is related to *Figure 7L,M*.

model in which mechanical force production via regulation of pMLC levels could play a central role in lung branching during development. These key findings establish a new conceptual framework for understanding how changes in local epithelial properties mediated by YAP drive lung branching morphogenesis. Major steps forward include studies that integrate dynamic changes in cellular properties in real time and investigations aimed to reveal how YAP senses changes in the external environment and the molecular mechanisms by which YAP outputs modify cellular behavior. These endeavors would unveil how Hippo signaling executes crucial steps of lung development. We anticipate that these results will also inform us how aberrant Hippo signaling leads to lung pathology.

Control of tissue tension by YAP has been described in medaka embryos (*Porazinski et al., 2015*). Interestingly, ARHGAP18 was reported to be an effector of YAP in that study. Moreover, the Hippo pathway can regulate F-actin accumulation in Drosophila (*Gaspar and Tapon, 2014*) and YAP regulates genes that control cell cycle progression, F-actin polymerization and actin cytoskeleton remodeling in the mammalian heart (*Morikawa et al., 2015*). These genes were proposed to function in cell migration in the heart.

By contrast, our work uncovers YAP targets that may regulate phosphorylation of myosin light chain and consequently mechanical force production in the lung. The actin cytoskeleton itself does not generate mechanical force. Mechanical force production is mediated by contraction of myosin in the actomyosin cytoskeleton, a network composed of myosin associated with F-actin bundles. Thus, our studies offer new insight into how YAP controls patterning of mammalian organs via cell number and mechanical force. It is unclear in which steps during lung branching these two cell properties are required and how they impact the morphogenetic movement. Nor is it clear what the functional consequence is when either cell number or mechanical force is disrupted. For instance, would a reduction in epithelial cell number lead to a smaller sized lung with grossly correct patterning? Answering these important questions would rely on additional genetic manipulations in mice. These studies will reveal the interplay between changes in cell number and acquisition of cellular properties, a longstanding question in tissue patterning.

In this study, we focus on how epithelial cell properties regulated by YAP contribute to lung branching. Given the complex interactions between the epithelium and mesenchyme during lung development, it is conceivable that coordination between these two compartments is critical for stereotypical branching. Indeed, classical transplantation studies have revealed the inductive ability of the mesenchyme in epithelial branching. Interestingly, recent studies suggest the involvement of mesenchymal smooth muscle in epithelial branching (*Kim et al., 2015*). How mesenchymal signals impinge on YAP activity is unclear. A better understanding of the molecular events in both the epithelial and mesenchymal compartments during branching is required to unveil the reciprocal interactions between these two compartments and the role that Hippo signaling plays in this process. We also anticipate that a combination of genetic studies, *ex vivo* organ explants and pharmacological and molecular manipulations would complement each other and bring us a more complete picture of how branching is executed at the molecular level.

Our work reveals a complex signaling network by which YAP regulates pMLC levels and cortical contractility. In addition to inducing *Arhgef17*, YAP also activates multiple pMLC regulators. This would ensure a dynamic control of pMLC levels in a temporally and spatially specific manner required for mechanical force production and lung branching. It is interesting to note that YAP also

regulates the expression of other regulators of the actomyosin cytoskeleton. While many regulators of the cytoskeleton are known to control Hippo signaling, our finding reveals a novel feedback control of the actomyosin cytoskeleton by YAP. This could function to fine-tune mechanical force production. Uncovering the *in vivo* function of YAP targets that we have identified during lung development would increase our understanding of how YAP controls cellular contractility.

We reported mosaic patterns of nuclear YAP distribution in lung epithelium. pYAP levels also vary significantly from cell to cell in both the proximal and distal airways, and a lower pYAP level is usually associated with nuclear localization of YAP. This suggests that YAP is dynamically shuttling in and out of the nucleus along the entire lung epithelium. On the average, 30–50% proximal and 10–28% distal epithelial cells contain cytoplasmic YAP, which is presumed to be inactive. Lung branching takes place at specific locations in the lung (*Metzger et al., 2008*). It is likely epithelial cells that undergo active branching need to maintain active YAP signaling. By contrast, epithelial cells not actively branching may have inactive YAP through YAP phosphorylation. We speculate that this could serve to facilitate directional epithelial movement and new bud formation. How nuclear YAP is activated and maintained in lung epithelial cells requires future investigations. Lung epithelial cells with scanty nuclear YAP usually stain positive for cytoplasmic YAP. If cytoplasmic YAP is inactive as postulated, we anticipate that deletion of YAP in these cells should not have significant functional consequences. Namely, deletion of cytoplasmic YAP by Cre is not expected to exert non-cell autonomous effects on neighboring cells that contain nuclear YAP. As an extension of these ideas, we propose that pMLC is only produced and/or maintained at active sites of branching. This would allow mechanical force production in a regional manner and induce corresponding morphological changes. Consistent with this idea, pMLC was shown to exhibit dynamical patterns in the epithelial membrane during branching morphogenesis (*Schnatwinkel and Niswander, 2013*). In the absence of YAP, disruption of pMLC production leads to a failure of new bud formation. One possibility is that new bud formation requires increased mechanical tension at the apical surface for its expansion and protrusion to produce new buds, while less tension is present in the interbud region. Testing these models relies on refined genetic manipulation coupled with cellular manipulation, real-time imaging and theoretical modeling.

Loss of YAP induced by *Shh^Cre* resulted in global cystic formation. It is interesting to note that the trachea and main stem bronchi appears to be spared. Moreover, most cysts also expressed SOX9. This could be due to a failure of *Sox9*-expressing progenitors to generate a sufficient number of cells that become the SOX2$^+$ conducting airway (*Alanis et al., 2014*). It is also possible that a fundamental difference in the mechanical properties of lung epithelial cells exists in different regions of the lung. In this scenario, cyst formation due to loss of mechanical force may be prone to occur at the distal tips of the lung. By contrast, lung epithelial cells in the proximal part of the airway may receive mechanical restraints from neighboring cells and cyst formation is less frequent.

How YAP integrates signals from the environment and orchestrates changes in local cell properties of the lung epithelium is unknown. Given that major signaling pathways control key aspects of lung branching, it is likely that these signals funnel through YAP. Interestingly, a previous study reported distal cyst formation in *Frizzled 2* (*Fzd2*)-deficient lungs (*Kadzik et al., 2014*). Loss of *Fzd2* also resulted in reduced apical pMLC expression (*Kadzik et al., 2014*). These findings led to a model in which non-canonical Wnt signaling through its receptor Fzd2 regulates epithelial cell behavior necessary for lung branching. This raises the interesting possibility that Wnt signaling could impinge upon YAP during branching morphogenesis. The relationship between Wnt and YAP is controversial in the literature. Future genetic studies in mice are required to clarify the physiological function of pathway interactions *in vivo*.

Our findings differ in several ways from a previous report in which localized YAP activity was detected in the lung (*Mahoney et al., 2014*). In this model, a nuclear-cytoplasmic shift of YAP protein marks the 'transition zone', a small proximal epithelial domain (SOX2$^+$) abutting the distal SOX9$^+$ epithelial compartment (*Mahoney et al., 2014*). Namely, nuclear YAP is present in cells in the distal airway and the transition zone, while cytoplasmic YAP is detected in the proximal airway. We found that active nuclear YAP is distributed along the entire airway epithelium. It is unclear why our results are different from the published work (*Mahoney et al., 2014*). We speculate that tissue processing and the staining procedure (such as tyramide signal amplification) may have played a role since they could unmask YAP antigens.

Moreover, nuclear YAP in SOX2$^+$ cells at the 'transition zone' was proposed in the published literature (*Mahoney et al., 2014*) to directly induce *Sox2* expression. This enables the initiation of the progenitor cell program that forms the airways during branching morphogenesis. The global cystic lung phenotype in *Yap*$^{f/f}$; *Shh*$^{Cre/+}$ mice was attributed to loss of a *TGFβ*-mediated proximal-distal program due to selective disruption of a YAP-SOX2 axis in the transition zone. While attractive, this model seems to be at odds with studies in other tissues where Hippo signaling exerts a general effect on all cells. In this study, we found no evidence of localized YAP activity to the transition zone. Importantly, if the lung phenotypes in *Yap* mutants can only be initiated at a specific location in the lung epithelium (such as the 'transition zone' connecting SOX2$^+$ and SOX9$^+$ populations), we do not anticipate lung cysts to form in *Yap*$^{f/f}$; *Sox9*$^{Cre/+}$ or *Yap*$^{f/f}$; *spc*$^{Cre/+}$ lungs since high levels of Cre activity are not present in the SOX2$^+$YAP$^{nuclear}$ transition zone in these mice. In fact, we observed distal cysts in both *Yap*$^{f/f}$; *Sox9*$^{Cre/+}$ and *Yap*$^{f/f}$; *spc*$^{Cre/+}$ lungs (*Figure 3*). These findings contradict the predictions of localized YAP activity in the transition zone and instead support our model of a general effect of YAP on epithelial cell properties. Taken together, our studies form the basis of future mechanistic studies to understand how modulation of local epithelial properties by major signaling pathways controls cell number, morphogenetic movement and pattern formation.

# Materials and methods

## Animal husbandry

*Shh*$^{Cre}$ [B6.Cg-*Shh*$^{tm1(EGFP/cre)Cjt}$/J; RRID:IMSR_JAX:005622], *Nkx2.1*$^{Cre}$ [C57BL/6J-*Tg (Nkx2–1-cre) 2Sand*/J] and *ROSA26*$^{mTmG}$ [*Gt(ROSA)26Sor*$^{tm4(ACTB-tdTomato,-EGFP)Luo}$/J; RRID: IMSR_JAX:007576] mice were obtained from Jackson Laboratory (Bar Harbor, ME, USA). *spc*$^{Cre}$ (*Sftpc-Cre*) mice [*Tg (Sftpc-cre)1Blh*; MGI:3574949] were given by Dr. Brigid Hogan and *Sox9*$^{Cre}$ [*Sox9t*$^{m3(Cre)Crm}$; MGI: 3608931] mice by Dr. Benoit de Crombrugghe. The *Yap* floxed allele [MGI: 5446483] was provided by Dr. Eric Olson. The Institutional Animal Care and Use Committee (IACUC) at the University of California, San Francisco, approved all experiments performed in this study.

Matings were set up to obtain *Yap*$^{f/f}$; *Shh*$^{Cre/+}$ mice, which are also called *Yap* mutants in this study since their lung phenotypes likely represent a complete loss of YAP activity in the lung epithelium. *Yap*$^{f/f}$; *Shh*$^{Cre/+}$ embryos were visibly smaller than their wild-type littermates and all of them had a short curly tail, which could be used to identify *Yap*$^{f/f}$; *Shh*$^{Cre/+}$ embryos by their outer appearance. Of note, *Yap*$^{f/f}$; *Shh*$^{Cre}$/*Cre* mice (carrying two copies of *Shh*$^{Cre}$) exhibited the classical Hedgehog defects due to disruption of the *Shh* locus and died at various time points during embryonic development (*Chiang et al., 1996*).

The efficiency of conversion of a conditional allele into a null allele by Cre recombinase varies for different conditional alleles. The *ROSA26*$^{mTmG}$ reporter is a very sensitive readout of Cre activity. In mice that carry *Sox9*$^{Cre/+}$; *ROSA26*$^{mTmG/+}$, *spc*$^{Cre/+}$; *ROSA26*$^{mTmG/+}$ or *Nkx2.1*$^{Cre/+}$; *ROSA26*$^{mTmG/+}$, most epithelial cells were labeled by GFP (*Song et al., 2012*). By contrast, Cre activity from these three lines was not sufficient to convert a conditional allele of *Yap* into a null allele except in the distal airway for *Sox9*$^{Cre}$ and *spc*$^{Cre}$ and in the upper lobe for *Nkx2.1*$^{Cre}$. Consistent with this notion, while Nkx2.1$^{Cre}$ activity was widely distributed along the lung epithelium as assessed by the R26R reporter (*Soriano, 1999*), Nkx2.1$^{Cre}$ activated *β*-catenin only in limited areas of the lung epithelium (*Li et al., 2009a*). A major factor that affects the efficiency of Cre excision is the distance between the two LoxP sites. We noticed that the distance between the two LoxP sites in the *Yap*$^f$ allele we used is approximately 1 kb. This could have contributed to the lower efficiency of conversion of a *Yap*$^f$ allele to a null allele by Cre lines in comparison with the reporter alleles.

## Histology and immunofluorescence

Mouse embryos were harvested at indicated time points, and the embryos or dissected lungs were fixed in 4% paraformaldehyde (PFA) in PBS for 1–2 hr at 4°C. Embryos or lungs were embedded in paraffin wax and sectioned at 6 µm or embedded in OCT and sectioned at 10 µm. Histological analysis was performed as reported (*Song et al., 2012*; *Lin et al., 2012*).

Histology and immunohistochemistry was performed following standard procedures. The following primary antibodies were used: rabbit anti-NKX2.1 (1:100; Epitomics (Burlingame, CA, USA) #2044–1; RRID:AB_1267367), mouse anti-p63 (1:100; Santa Cruz Biotechnology (Dallas, TX, USA)

#sc8431; RRID:AB_628091), goat anti-Clara cell 10 kDa protein (CC10) (S20) (1:200; Santa Cruz Biotechnology #sc-9773; RRID:AB_2183391), mouse anti-acetylated (Ac)-tubulin (1:1000; Sigma-Aldrich (St. Louis, MO, USA) #T6793; RRID:AB_477585), rabbit anti-prosurfactant protein C (proSPC) (1:400; MilliporeEMD Millipore (Billerica, MA, USA) #AB3786; RRID:AB_91588), hamster anti-T1α (1:200; Developmental Studies Hybridoma Bank (Iowa City, IA, USA) #8.1.1; RRID:AB_531893), rat anti-E-cadherin (1:500; Life Technologies (Carlsbad, CA, USA) #13–1900; RRID:AB_2533005), rabbit anti-SOX2 (1:50; Abcam (Cambridge, MA, USA) #ab97959; RRID:AB_2341193), goat anti-SOX9 (1:50; R&D Systems (Minneapolis, ME, USA) #AF3075; RRID:AB_2194160), rabbit anti-pMLC (S19) (1:50; Cell Signaling Technology (Danvers, MA, USA) #3671S; RRID:AB_330248), goat anti-CTGF (1:100; Santa Cruz Biotechnology #sc-14939; RRID:AB_638805), mouse anti-YAP (1:100; Santa Cruz Biotechnology #sc-101199; lot number F0214; RRID:AB_1131430), rabbit anti-YAP (1:100; Novus Biologicals (Littleton, CO, USA) #NB110–58358; RRID:AB_922796), rabbit anti-phospho-YAP (1:100; Cell Signaling #4911S; RRID:AB_2218913), rabbit anti-Cre (1:1000; Millipore #69050–3; RRID:AB_11212994) and mouse anti-Cre (1:1000; Millipore #MAB3120; RRID:AB_2085748). F-actin was stained with rhodamine-conjugated phalloidin (1:200; Sigma). Secondary antibodies and conjugates used were donkey anti-rabbit Alexa Fluor 488 or 594 (1:1000; Life Technologies), donkey anti-goat Alexa Fluor 488, 594, or 647 (1:1000; Life Technologies), donkey anti-mouse Alexa Fluor 594 (1:1000; Life Technologies), and DAPI (1:10,000; Sigma).

For biotinylated secondary antibodies (goat anti-hamster, 1:1000; goat anti-rabbit, 1:1000; donkey anti-goat, 1:1000; donkey anti-rat, 1:1000; and horse anti-mouse, 1:1000; Jackson ImmunoResearch Laboratories [West Grove, PA, USA]), the signal was detected using streptavidin-conjugated Alexa Fluor 488, 594, or 647 (1:1000; Life Technologies) or HRP-conjugated streptavidin (1:1000; Perkin-Elmer [Boston, MA, USA]) in combination with either the chromogenic substrate DAB (Vector Laboratories [Burlingame, CA, USA]) or fluorogenic substrate Alexa Fluor 488 tyramide (1:200, TSA kit; Perkin-Elmer).

For immunofluorescence of YAP protein, paraffin sections were deparaffinized and antigen retrieval was performed in sodium citrate solution in a microwave oven. The slides were permeabilized in 0.5% Triton in PBS for 10 min and incubated with preblock buffer (3% BSA/0.1% Triton/PBS) for 1 hr. The samples were incubated at 4°C overnight with primary antibodies diluted (1:100) in preblock buffer. After washes with PBS, the samples were incubated with biotinylated anti-mouse or anti-rabbit secondary antibodies diluted (1:1000) in preblock buffer for 1 hr at room temperature. The signals were detected using HRP-conjugated streptavidin (1:500; Vector Laboratories) followed by tyramide signal amplification (TSA) for 30 s (1:100, TSA kit; Perkin Elmer). The following YAP antibodies were used: mouse anti-YAP (1:100; Santa Cruz Biotechnology #sc-101199; lot #F0214; RRID: AB_1131430) and rabbit anti-YAP (1:100; Novus Biologicals #NB110–58358; RRID:AB_922796). Both YAP antibodies have been widely used in the literature and similar results were obtained from both antibodies.

For BrdU or EdU incorporation, mice were injected with 1–2 mg of BrdU or EdU solution and embryos or lungs were collected 1 hr following BrdU or EdU injection. BrdU staining was performed using the BrdU Staining Kit (Life Technologies). EdU staining was performed using the Click-iT EdU Alexa Fluor 488 Imaging Kit (Life Technologies).

For TUNEL analysis, paraffin sections were deparaffinized and antigen retrieved. The sections were permeabilized in 0.5% Triton-X100/PBS for 15 min, blocked in 3% BSA for 1 hr at room temperature and incubated with rabbit anti-NKX2.1 antibodies (1:100; Epitomics #2044–1; RRID:AB_1267367) at 4°C overnight. Anti-rabbit Alexa Fluor 594 secondary antibodies (Molecular Probes) were added to the TUNEL reaction mix, which was prepared by diluting one part Enzyme Solution in nine parts Label Solution from the In Situ Cell Death Detection Kit (Roche Applied Science [Penzberg, Germany]). The sections were incubated with the secondary antibody/TUNEL reaction mix for 1 hr at 37°C, washed in PBS three times, incubated with DAPI for 5 min and mounted in Vectashield (Vector Laboratories) for microscopy.

Confocal images were captured on a Leica Microsystems (Wetzlar, Germany) laser-scanning confocal microscope. Adjustment of red/green/blue histograms and channel merges were performed using LAS AF Lite (Leica).

## Whole-mount immunofluorescence

Whole-mount immunofluorescence of lungs was performed mainly following a previously described protocol (*Metzger et al., 2008*) but without tyramide amplification. Briefly, lungs from wild-type and *Yap^{f/f}; Shh^{Cre/+}* embryos at 11.5–14.5 *dpc* were dissected and dehydrated in graded methanols (25%, 50%, 75%, 100%) and stored at –20°C. On the first day of the experiment, lungs were incubated in 5% $H_2O_2$/methanol for 5 hr and then rehydrated through graded methanols diluted in 0.1% Tween-20/PBS. After blocking in preblock buffer (1.5% BSA/0.5% Triton X-100/PBS) for 1 hr twice, the samples were incubated with primary antibodies at 4°C overnight. The primary antibodies used were: rabbit anti-phospho MLC (S19) (1:100; Cell Signaling Technology #3671S; RRID:AB_330248) and rat anti-E-cadherin (1:500; Life Technologies #13–1900; RRID:AB_2533005).

On the second day, the samples were washed in preblock buffer at 4°C for 1 hr five times. This was followed by overnight incubation with secondary antibody (donkey anti-rabbit 488 and donkey anti-rat 594) diluted 1:250 in preblock buffer at 4°C. On the third day, the samples were washed with preblock buffer for 1 hr five times at 4°C and then with 0.1% Triton-PBS for 1 hr twice at room temperature. Finally, the samples were mounted in Vectashield (Vector Laboratories) and subjected to two-photon microscopy analysis.

Two-photon microscopy was performed using an upright LSM 7 MP laser-scanning microscope (Carl Zeiss [Oberkochen, Germany]) outfitted with a W Plan-Apochromat water-immersion 20× objective (numerical aperture, 1.0) and ZEN 2009 software (Carl Zeiss).

## Time-lapse fluorescence microscopy

Mouse lungs from control and *Yap^{f/f}; Shh^{Cre/+}; ROSA26^{mTmG/+}* embryos at 12.5 *dpc* were dissected and placed on polycarbonate nuclepore membranes (Millipore) with lungs floating in DMEM:F12 (1:1) medium (Gibco) containing 0.5% FBS (Gemini Bio-Products [West Sacramento, CA, USA]) and 1% penicillin–streptomycin (Sigma). A 24-well plate that contained the nuclepore membranes was then put in a stage-top environmental chamber for time-lapse fluorescence microscopy. The chamber maintained humidity, temperature at 37°C, and $CO_2$ at 5%. For live imaging experiments, time-lapse images were captured with a wide-field epifluorescence microscope on an inverted microscope (Eclipse Ti-E, Nikon [Chiyoda, Japan]) equipped with Sutter Lambda LS Arc Lamp, a Perfect Focus system (Nikon) and a 4×/0.72 Plan Apo objective lens. Lungs were imaged every 30 min for 12 hr. All images were analyzed with NIS-Elements Advanced Research software (Nikon).

## Mechanical force measurement

Lungs from control and *Yap^{f/f}; Shh^{Cre/+}; ROSA26^{mTmG/+}* embryos at 12.5 *dpc* were collected in cold PBS and mounted onto concavity slides with coverslips. Epithelial cells were imaged through their expression of GFP (from the *ROSA26^{mTmG/+}* allele) with a 40× oil objective (NA = 1.2, Zeiss). A 10 µm span of epithelial cells in the left lobe was targeted for 2 s by a Chameleon 915 nm two-photon laser (~700 mW) mounted on a custom-built Zeiss upright fluorescence microscope. Immediately following ablation, time-series images of epithelial cells were taken at the frame rate of 1 s for up to 50 s. The movements of the epithelial cells from 0 to 5 s were tracked using the manual tracking plug-in of ImageJ (NIH).

## Western blot analysis

Embryonic lung tissues were homogenized in RIPA buffer supplemented with Complete Protease Inhibitor Cocktail tablets (Roche) and phosSTOP Phosphatase Inhibitor Cocktail tablets (Roche). The lysates were cleared and analyzed by Western blot as previously described (*Lin et al., 2012*).

The following primary antibodies were used: rabbit anti-YAP (1:500; Cell Signaling Technology #4912S; RRID:AB_2218911), rabbit anti-MLC (1:500; Cell Signaling Technology #3672; RRID:AB_10692513), rabbit anti-pMLC (S19) (1:500; Cell Signaling Technology #3671S; RRID:AB_330248), mouse anti-RhoA (1:2000; Santa Cruz Biotechnology #sc-418; RRID:AB_628218), goat anti-ROCK1 (1:100; Santa Cruz Biotechnology #sc-6055; RRID:AB_2182155) and mouse anti-ROCK2 (1:2000; BD Biosciences #610623; RRID:AB_397955).

## RNA-Seq

RNA was extracted from the lungs of *Yap^{f/f}; Shh^{Cre/+}* embryos and their wild-type littermates at 12.5 and 14.5 *dpc* using TRIzol (Life Technologies) and the RNeasy Kit (Qiagen [Hilden, Germany]). RNA quality was evaluated using the Agilent 2100 Bioanalyzer (Agilent Technologies [Santa Clara, CA, USA]).

Paired-end libraries were prepared using the SureSelect Strand-Specific RNA Library Prep kit (Agilent Technologies). Multiplexed sequencing was run in a HiSeq2000 or HiSeq4000 sequencer (Illumina [San Diego, CA, USA]). Read alignment and differentially expressed genes were analyzed by the Maverix Biomics (San Mateo, CA, USA) Analytic Platform (Maverix). Functional enrichment analysis was performed using Ingenuity Pathway Analysis software (version 7.1). Datasets are deposited on the Gene Expression Omnibus database under the accession number GSE93339 (GEO, https://www.ncbi.nlm.nih.gov/geo/query/acc.cgi?acc=GSE93339).

## Chromatin immunoprecipitation (ChIP)-Seq

Fifty milligrams of embryonic lung tissues were minced into small pieces, crosslinked with 2 mM disuccinimidyl glutarate (DSG) (Thermo Fisher Scientific [Waltham, MA, USA]) for 40 min, fixed with 1% formaldehyde for 10 min and quenched with 0.125 M glycine for 5 min. Tissues were resuspended in 300 µl lysis buffer (50 mM Tris-HCl, pH 8.1, 10 mM EDTA, 1.0% SDS) supplemented with protease inhibitors. Chromatin was sheared to a range of 100–400 base pairs (bp) in size by sonication for 30 min using a Bioruptor Sonicator (Diagenode [Denville, NJ, USA]). The sheared chromatin was diluted 1:10 with ChIP dilution buffer (20 mM Tris-HCl, pH 8.1, 150 mM NaCl, 2 mM EDTA, 1.0% Triton X-100) and pre-cleared with 110 µl Protein A/G PLUS agarose (Santa Cruz Biotechnology) for 2 hr at 4°C. The pre-cleared chromatin was divided into two halves; each half was incubated with either 2 µg rabbit anti-YAP antibody (Novus Biologicals #NB110–58358; RRID:AB_922796) or 2 µg rabbit normal IgG (Santa Cruz Biotechnology) at 4°C overnight. For ChIP-Seq of histone modifications, each half of the pre-cleared chromatin was incubated with 2 µg rabbit anti-H3K4me1 antibody (Abcam #ab8895, RRID:AB_306847) or 2 µg rabbit anti-H3K4me3 antibody (Abcam #ab8580, RRID: AB_306649). Forty-five microliters Protein A/G PLUS agarose were added to each sample, which was rocked at 4°C for 1 hr. The immunoprecipitates were washed sequentially with TSE I buffer (150 mM NaCl, 20 mM Tris-HCl, pH 8.1, 0.1% SDS, 1.0% Triton X-100), TSE II buffer (500 mM NaCl, 20 mM Tris-HCl, pH 8.1, 0.1% SDS, 2 mM EDTA, 1.0% Triton X-100), buffer 3 (250 mM LiCl, 10 mM Tris-HCl, pH 8.1, 1 mM EDTA, 1.0% NP-40, 1.0% Triton X-100) and TE buffer (10 mM Tris-HCl, pH 8.1, 1 mM EDTA). Immunoprecipitates were eluted twice in 250 µl elution buffer (100 mM NaHCO₂, 1.0% SDS) at room temperature for 15 min. Crosslinking was reversed by incubating samples at 65°C for 4 hr and the samples were then treated with proteinase K at 45°C for 1 hr. DNA was extracted using Qiagen PCR-purification Kits. Two biological replicates using YAP and IgG antibodies were obtained.

Illumina libraries were constructed from the ChIP DNA and sequenced single-ended on a HiSeq2500 (Illumina) at the UC Davis genome center. ChIP-Seq reads were demultiplexed and aligned to the mouse genome (mm10) using Bowtie with default settings (*Langmead and Salzberg, 2012*). The resulting SAM files were converted to the BAM format by Samtools (*Li et al., 2009b*) for peak calling with MACS2 (version 2.0.10) (*Zhang et al., 2008*). For peak calling, MACS2 extended the reads to 300 bp and kept only peaks where FDR ≤ 0.05. Bedtools was used to obtain reproducible peaks between two wild-type replicates (*Quinlan et al., 2010*). *De novo* motif discovery was performed in 100 bp windows centered at the peak summits with MEME (*Bailey et al., 2009*).

## Genome-wide identification of conserved TEAD-binding sites in gene promoters

Searching for putative TEAD-binding sites (GGAATG) on mouse gene promoters (−500 to 0; transcription start as +1) was performed using the Regulatory Sequence Analysis Tools (RSAT) (http://rsat.ulb.ac.be/rsat/) and implemented with the pattern matching tool. The conservation of the *cis*-regulatory elements identified was assessed using the 'Conservation' track from the UCSC browser (http://genome.ucsc.edu/).

## ChIP-qPCR

The procedure of ChIP using YAP antibody and rabbit IgG was similar to that described above for ChIP-Seq. The procedure of ChIP using TEAD1 antibody has the following modifications: (1) 50 mg of embryonic lung tissues were fixed in 1% formaldehyde for 15 min and the DSG crosslinking step was skipped; (2) after pre-clearing, each half of the chromatin was incubated with either mouse anti-TEAD1 (TEF1) antibody (BD Biosciences [San Jose, CA, USA] #610922; RRID:AB_398237) or mouse normal IgG (Santa Cruz Biotechnology).

The primers for qPCR were mouse *β-actin* promoter (forward, 5′-AGAAGGACTCCTATGTGGG TGA-3′; reverse, 5′-ACTGACCTGGGTCATCTTTTC-3′), mouse *Ccne1* promoter (forward, 5′-CC TCCCACTTCTCTTTCTTCTTTC-3′; reverse, 5′-TTATCTTAATACAATGGTAGTCTTCAAGC-3′), mouse *Arhgef17* promoter (forward, 5′-AGGAGGCAATGGAGGAGG-3′; reverse, 5′-GGCGGATGGTTTACA TTCTTG-3′), mouse *S1pr2* promoter (forward, 5′-CACTATAGGAAGCTGAAGCCG-3′; reverse, 5′-C TGATAAGGAGCTGGAGAGTG-3′), mouse *Bcam* promoter (forward, 5′-AATCCAAGGAATG TCACCCC-3′; reverse, 5′-CCCACTTCTCCTCCCCTC-3′), mouse *Nuak2* promoter (containing the TEAD-binding site) (forward, 5′-TCCCACAGCGTTTATTCCC-3′; reverse, 5′-GGGCATTCCAAGCA TTCTTG-3′), mouse *Nuak2* promoter (forward, 5′-ATCCTAAAGACTGGCACTTCG-3′; reverse, 5′-CA TTGGTTCACCCTCTCCTG-3′), mouse *Amotl2* promoter (forward, 5′-AACTCTCACATTCCTGGCA TAG-3′; reverse, 5′-CAGTCAGCAACGGAGGTG-3′), mouse *Prkci* promoter (forward, 5′-AGGCTGG TGGGTTCTGTTCC-3′; reverse, 5′-GCTCCCAAGGCCGCATTC-3′), mouse *Foxp2* promoter (forward, 5′-CAGGAATCTGCGACAGAGAC-3′; reverse, 5′-TTACTTCAGAGCTGGTGTCAC-3′), mouse *Ntn2l* promoter (forward, 5′-GGCTGTAAGGCTGAGCTG-3′; reverse, 5′-AGAAGCAGATTCAGACACAGG-3′), mouse *Itgb6* promoter (forward, 5′-GTGAGTTTAAACCTAAGCTGCC-3′; reverse, 5′-CATAG TTGAGCACATACCCAGG-3′), mouse *Fbln7* promoter (forward, 5′-GCATTCCAGGCTCCACAG-3′; reverse, 5′-GCTTCCAAGGCCACTAGTC-3′), mouse *Myl12b* promoter (forward, 5′-CTCGGGAA TGCGGAAGG-3′; reverse, 5′-GAGTACATATTCCCCAGCTCAC-3′).

## qPCR analysis

Total RNA was extracted from lung tissues or cultured cells using TRIzol (Life Technologies) and subsequently reverse-transcribed using the Maxima First Strand cDNA Synthesis Kit (Thermo Scientific). Quantitative PCR (qPCR) was carried out on the ABI Prism 7900HT Sequence Detection System.

The following primers for mouse genes were used: *Gapdh* (forward, 5′-AGGTTGTCTCCTGCGAC TTCA-3′; reverse, 5′-CCAGGAAATGAGCTTGACAAAGTT-3′), *Ctgf* (forward, 5′-CTCCACCCGAG TTACCAATG-3′; reverse, 5′-TGGCGATTTTAGGTGTCCG-3′), *Shh* (forward, 5′-CAGGTTTCGAC TGGGTCTACTATG 3′; reverse, 5′-TTTGGCCGCCACGGAGTT-3′), *Ptch1* (forward, 5′-TGCTGTGCC TGTGGTCATCCTGATT-3′; reverse, 5′-CAGAGCGAGCATAGCCCTGTGGTTC-3′), *Ptch2* (forward, 5′- GGCACTCACATCCGTCAACAAC-3′; reverse, 5′-GAAGACGAGCATTACCGCTGCA-3′), *Gli1* (forward, 5′-CCCATAGGGTCTCGGGGTCTCAAAC-3′; reverse, 5′-GGAGGACCTGCGGCTGACTGTG TAA-3′), *Hhip1* (forward, 5′-CAACCAGGAACGGTGGGCTATT-3′; reverse, 5′-TCTGCGACTTCCA GAAACACCC-3′), *Fgf10* (forward, 5′-ACCAAGAATGAAGACTGTCCG-3′; reverse, 5′-TTTGAGCCA TAGAGTTTCCCC-3′), *Pdgfa* (forward, 5′-GCAGTTGCCTTACGACTCCAGA-3′; reverse, 5′-GG TTTGAGCATCTTCACAGCCAC-3′), *Pdgfra* (forward, 5′-TGCAGTTGCCTTACGACTCCAGAT-3′; reverse, 5′-AGCCACCTTCATTACAGGTTGGGA-3′), *Wnt7b* (forward, 5′-TTCTCGTCGCTTTGTGGA TGCC-3′; reverse, 5′–CACCGTGACACTTACATTCCAGC-3′), *Ccnd1* (forward, 5′-GCAGAAGGAGA TTGTGCCATCC-3; reverse, 5′-AGGAAGCGGTCCAGGTAGTTCA-3′), *Ccne*1 (forward, 5′-AAGCCC TCTGACCATTGTGTCC-3′; reverse, 5′-CTAAGCAGCCAACATCCAGGAC-3′), *Arhgef17* (forward, 5′-TTCTATGTTCAACCCCACCG-3′; reverse, 5′-GAAGTCCTCAGAGCCATCAC-3′), *S1pr2* (forward, 5′-CCAACAGTCTCCAAAACCAAC-3′; reverse, 5′-GAGTATAAGCCGCCCATGG-3′), *Bcam* (forward, 5′-AGAGTGGAGGATTACGATGCCG-3′; reverse, 5′-TGCTGTTCAGGAATACGAAGAGC-3′), *Nuak2* (forward, 5′-CTGGTGAAGCAAATCAGTAACGG-3′; reverse, 5′-CCACCAATGACTGGCTACATCC-3′), *Amotl2* (forward, 5′-AACTCTCACATTCCTGGCATAG-3′; reverse, 5′-CAGTCAGCAACGGAGG TG-3′), *Prkci* (forward, 5′-AGGCTGGTGGGTTCTGTTCC-3′; reverse, 5′-GCTCCCAAGGCCGCATTC-3′), *Myl12b* (forward, 5′-CTCGGGAATGCGGAAGG-3′; reverse, 5′-GAGTACATATTCCCCAGC TCAC-3′), *Rai14* (forward, 5′-GCGGAGAACATTGACAACTCGG-3′; reverse, 5′-CTTGTGTTCGCA GAGGAGCTGT-3′), *Itgb6* (forward, 5′-ACTCATTCCTGGAGCAACCGTG-3′; reverse, 5′-GCTG TGAAAGACAGGTTGAGTCC-3′), *Fbln7* (forward, 5′-ATGGTAGCTGGACAGGAGAGCA-3′; reverse,

5′-ATGCTGACAGCGGTTCCCAGTT-3′), *Cldn4* (forward, 5′-GTAGCAACGACAAGCCCTAC-3′; reverse, 5′-AGGCAATGTGGACAGAGTG-3′).

## Acknowledgements

We thank Rhodora Gacayan and Peter Johnson for technical assistance, Walter Eckalbar for his help with ChIP-Seq analysis and Ross Metzger for critical reading of the manuscript. Some data for this study were acquired at the Nikon Imaging Center at CVRI. We thank Chris Allen and Xin-Zi Tang for their help with two-photon microscopy and Yuh-Nung Jan for the use of his two-photon microscope for laser ablation. This work was supported by grants (U01 HL111054 and R01 HL115207) from the National Institutes of Health to PTC.

## Additional information

### Funding

| Funder | Grant reference number | Author |
|---|---|---|
| Foundation for the National Institutes of Health | U01 HL111054 | Pao-Tien Chuang |
| Foundation for the National Institutes of Health | R01 HL115207 | Pao-Tien Chuang |

The funders had no role in study design, data collection and interpretation, or the decision to submit the work for publication.

### Author contributions

CL, EY, Conceptualization, Data curation, Formal analysis, Validation, Investigation, Methodology, Writing—review and editing; KZ, Data curation, Investigation; XJ, Data curation, Formal analysis, Methodology; SC, KT-P, Data curation, Formal analysis, Investigation; P-TC, Conceptualization, Data curation, Formal analysis, Supervision, Funding acquisition, Investigation, Methodology, Writing—original draft, Project administration, Writing—review and editing

### Author ORCIDs

Katherine Thompson-Peer, http://orcid.org/0000-0002-4200-3870
Pao-Tien Chuang, http://orcid.org/0000-0002-8961-8653

### Ethics

Animal experimentation: This study was performed in strict accordance with the recommendations in the Guide for the Care and Use of Laboratory Animals of the National Institutes of Health. The Institutional Animal Care and Use Committee (IACUC) at the University of California, San Francisco, approved all experiments performed in this study (protocol #AN109065-02).

## Additional files

### Major datasets

The following dataset was generated:

| Author(s) | Year | Dataset title | Dataset URL | Database, license, and accessibility information |
|---|---|---|---|---|
| Pao-Tien Chuang, Chuwen Lin | 2017 | Transcriptome analysis of wild-type and Yap-deficient embryonic lungs by RNA-Seq | https://www.ncbi.nlm.nih.gov/geo/query/acc.cgi?acc=GSE93339 | Publicly available at NCBI (accession no. GSE93339) |

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
