## [Decision Letter]

Thank you for submitting your article "YAP is essential for mechanical force production and epithelial cell proliferation during lung branching morphogenesis" for consideration by *eLife*. Your article has been favorably evaluated by Fiona Watt (Senior Editor) and three reviewers, one of whom is a member of our Board of Reviewing Editors. The following individual involved in review of your submission has agreed to reveal his identity: Tushar Desai (Reviewer #3).

The reviewers have discussed the reviews with one another and the Reviewing Editor has drafted this decision to help you prepare a revised submission.

Summary:

In their manuscript, Lin and colleagues investigate the role of the Hippo mediator, YAP, in the epithelium of the developing mouse lung. They show that nuclear (active) YAP is present in both proximal and distal compartments of the embryonic lung, and that targeted epithelial deletion of YAP early in lung development severely disrupts branching morphogenesis. Using mosaic epithelial deletion, they show that the cystic airway phenotype appears to be delimited to the regions where YAP was deleted, suggesting it acts locally to regulate branching. They then use RNA-Seq and ChIP-seq to identify differentially expressed genes and direct YAP target genes, respectively. Enrichment of cell cycle and cellular contractility genes prompted them to ask whether YAP controls cellular contractility, and they show that phospho-MLC is reduced in mutants. They then show a functional reduction in the speed of uninjured epithelial cell movement in mutant airways following neighbor ablation, consistent with a role for YAP in mechanical force production. Direct YAP target genes that activate RhoA, such as *Arhgef17*, are found to be reduced in mutants, supporting that YAP is upstream of mechanical force production.

This is an extensive study that reveals new information about the role of YAP during mouse lung development and puts forth an interesting model about YAP, which is itself regulated by mechanical tension, and might in turn regulate mechanical force generation.

Essential revisions:

1) Provide more convincing experimental data as to whether or not branch initiation, however perturbed, is abrogated in the YAP mutants. Specifically, analyze H&E sections scored for 'evaginations' (the putative 'branches' reported in the previous publication), as well as some live imaging of the mutant lungs in culture which could help delineate whether localized branches initiate or not.

2) Formulate more clearly a proposed model for how loss of mechanical force production in the epithelium translates into the observed cystic phenotype. Could this phenotype result without a 'branching' defect per se? Also, since the authors have not directly implicated mechanical force as the mechanism, they should temper the strength of their claim.

3) At the stages reported, a significant proportion of proximal (30-50%) and distal (10-28%) epithelial cells lack nuclear (active) YAP. The authors should discuss how this mosaicism factors into their proposed model. Do these cells stain for cytoplasmic YAP and, if so, could deletion of YAP in these cells be significant? If not, then does their failure to 'rescue' the YAP-deleted cells indicate a non-cell autonomous phenotype – the authors should clarify since they make a point of stating that the effects of YAP deletion are 'local' in the lung.

Suggested revisions:

The authors present an enormous amount of data characterizing various aspects of their mutant phenotypes. While this is an impressive amount of work, the way it is organized makes the manuscript difficult to follow. Certain parts of the figures are slightly redundant and could be omitted (e.g. Figure 1—figure supplements 3 and 4). On the other hand, the authors often refer to data that is not included to justify their conclusions (e.g. aPKC, β-catenin and α-catenin distribution), and should therefore be part of the manuscript.

---

## [Author Response]

*Essential revisions:*

*1) Provide more convincing experimental data as to whether or not branch initiation, however perturbed, is abrogated in the YAP mutants. Specifically, analyze H&E sections scored for 'evaginations' (the putative 'branches' reported in the previous publication), as well as some live imaging of the mutant lungs in culture which could help delineate whether localized branches initiate or not.*

As suggested by the reviewers, we have examined H&E sections as well as whole-mount images of control and *Yap*-deficient lungs. We found that at early stages of lung development, such as 11.5 and 12.5 *dpc*, limited lung branching occurred in the absence of YAP. Five lung buds could be discerned in *Yap*-deficient lungs at 11.5 *dpc*, although the extent of lung bud separation was not as complete as that in wild-type lungs (Figure 2; Figure 3). Limited branching continued at 12.5 *dpc* (Figure 2). Small “evaginations” could be found in *Yap*-deficient lungs (Figure 2) but they failed to generate new buds beyond 13.5 *dpc* (Figure 3).

This conclusion was supported by live imaging of *Yap*-deficient lungs using time-lapse microscopy and *ex vivo*lung explants. For instance, we examined lung explants at 12.5 *dpc*. At this stage, limited lung branching in *Yap*-deficient mutants would soon stop. We found that the existing lung buds developed multiple small “evaginations” over time but within the time frame of live imaging, they never progressed to generate new lung buds (Figure 2—figure supplement 1). This suggests that loss of YAP severely compromises branching after 12.5 *dpc* despite the presence of multiple small “evaginations”.

*2) Formulate more clearly a proposed model for how loss of mechanical force production in the epithelium translates into the observed cystic phenotype. Could this phenotype result without a 'branching' defect per se? Also, since the authors have not directly implicated mechanical force as the mechanism, they should temper the strength of their claim.*

In the Discussion section of the revised manuscript, we have formulated a more precise model for how loss of mechanical force production in the *Yap*-deficient lung epithelium could lead to the cystic phenotypes. In essence, we propose that regional YAP activation and mechanical force production may result in selective epithelial expansion and formation of new lung buds. Loss of YAP and mechanical force would compromise this process; limited omnidirectional outgrowth would lead to cyst formation.

We have toned down the claim of mechanical force in lung branching throughout the text of the revised manuscript. Our studies serve as the foundation for future investigation to fully understand the relationship between mechanical force production and lung branching.

*3) At the stages reported, a significant proportion of proximal (30-50%) and distal (10-28%) epithelial cells lack nuclear (active) YAP. The authors should discuss how this mosaicism factors into their proposed model. Do these cells stain for cytoplasmic YAP and, if so, could deletion of YAP in these cells be significant? If not, then does their failure to 'rescue' the YAP-deleted cells indicate a non-cell autonomous phenotype – the authors should clarify since they make a point of stating that the effects of YAP deletion are 'local' in the lung.*

We have included discussion on the mosaic pattern of nuclear YAP distribution. pYAP levels also vary significantly from cell to cell in both the proximal and distal airways and a lower pYAP level is usually associated with nuclear localization of YAP. This suggests that YAP is dynamically shuttling in and out of the nucleus along the entire lung epithelium. On the average, 30%-50% proximal and 10-28% distal epithelial cells contain cytoplasmic YAP, which is presumed to be inactive.

Lung branching takes place at specific locations in the lung. It is likely epithelial cells that undergo active branching need to maintain active YAP signaling. By contrast, epithelial cells not undergoing active branching may have inactive YAP through YAP phosphorylation. We speculate that this could serve to facilitate directional epithelial movement and new bud formation. How nuclear YAP is activated and maintained in lung epithelial cells requires future investigations.

Lung epithelial cells that do not have nuclear YAP usually stain positive for cytoplasmic YAP, but some cells have no YAP, presumably because YAP is degraded through the phosphorylation-induced degradation mechanism. If cytoplasmic YAP is inactive as postulated, we expect that deletion of YAP in these cells should not have significant functional consequences. In other words, deletion of cytoplasmic YAP by Cre is not expected to exert non-cell autonomous effects of neighboring cells that contain nuclear YAP. These points have been discussed in the revised manuscript.

*Suggested revisions:*

*The authors present an enormous amount of data characterizing various aspects of their mutant phenotypes. While this is an impressive amount of work, the way it is organized makes the manuscript difficult to follow. Certain parts of the figures are slightly redundant and could be omitted (e.g.* Figure 1—figure supplements 3 and 4*). On the other hand, the authors often refer to data that is not included to justify their conclusions (e.g. aPKC, β-catenin and α-catenin distribution), and should therefore be part of the manuscript.*

We have revised the manuscript per the reviewers’ suggestions. We have removed the original Figure 1—figure supplements 3 and 4. We have also added a new figure supplement (Figure 2—figure supplement 3) that includes data on the distribution of aPKC, β-catenin and α-catenin.